# Effect of different source terms and inflow direction in atmospheric boundary modeling over the complex terrain site of Perdigão

Kartik Venkatraman[1,2,*], Trond-Ola Hågbo[1,3,*], Sophia Buckingham[1], and Knut Erik Teigen Giljarhus[3]

[1]von Karman Institute for Fluid Dynamics, Waterloosesteenweg 72, B-1640 Sint-Genesius-Rode, Belgium
[2]Université de Sherbrooke, Canada
[3]University of Stavanger, Norway
[*]These authors contributed equally to this work.

**Correspondence:** kartik.venkatraman@vki.ac.be

**Abstract.** Assessing wind conditions in complex terrain requires Computational Fluid Dynamics (CFD) simulations incorporating an accurate parameterization of forest canopy effects and Coriolis effects. This study investigates how incorporating source terms such as the presence of trees and the Coriolis force can improve flow predictions. Furthermore, the study examines the impact of using different sets of atmospheric boundary layer inflow profiles, including idealized profiles with a logarithmic velocity profile, and a set of fully developed profiles from a pressure-driven precursor simulation. A three-dimensional steady Reynolds-averaged Navier-Stokes (RANS) equations model is set up using OpenFOAM to simulate the flow over a complex terrain site comprising two parallel ridges near Perdigão, Portugal. A 7.5 km $\times$ 7.5 km terrain of the Perdigão site is constructed from elevation data centered around a 100 m met-mast located on the Southwest ridge. A 30-min averaged stationary period is simulated, which corresponds to near-neutral conditions at met-mast Tower 20 located at the Southwest ridge. The period corresponds to the wind coming from Southwest at $231°$ at 100 m height above ground at Tower 20. Five case setups are simulated using a combination of different source terms, turbulence models and inflow profiles. The prediction capability of these models is analyzed for different groups of towers on the Southwest ridge and, on the towers further downstream inside the valley, and on the Northeast ridge. Including a canopy model improves predictions close to the ground for most of the towers on the Southwest ridge and inside the valley. Large uncertainties are seen in field measurement data inside the valley, which is a re-circulation zone and large prediction errors are seen in the wind velocity, wind direction and turbulent kinetic profiles for most of the models. The predictions on the Northeast ridge is dependent on the extent of re-circulation predicted inside the valley. The inflow wind direction plays an important role in wind profile predictions.

## 1 Introduction

Lack of terrain availability in flat terrain pushes wind-farm developers to look for alternatives along complex terrain sites. Flat terrain availability is becoming scarce and 70% of the Earth's surface is in complex terrain, which presents a large potential for wind energy harvesting. Winds in complex terrains are governed by the surface properties of the flow (land class/roughness length) and the local elevation such as hills, ridges and mountains (Emeis 2018). Local features such as ridges or canyons can also be advantageous for wind energy harvesting, due to the creation of local flow accelerations on top of ridges and flow chan-

neling through valleys. However, the wind fields depend on either the local pressure or temperature gradient. Such flows are also dominated by strong thermal stratification effects and are influenced by the presence of forest canopies. Complex terrain sites remain very challenging areas to consider for wind farm siting and require extensive validation of modeling tools in representative environments. Wind farm modeling for complex terrains requires a more advanced approach than commonly used cost-effective linearized models such as WAsP ( Jørgensen et al.,  2005) which cannot handle complex phenomena (i.e. flow separation) to ensure reliable and accurate results. Computational fluid dynamics (CFD) tools are increasingly used to predict flows over complex terrain sites to account for such phenomena and provide more accurate wind resource predictions (Blocken, 2014). However, improving numerical modeling tools for complex terrains demands accounting for various microscale phenomena such as flow re-circulation and the forest canopy effect. Such microscale flow features significantly impact the local wind resource assessment and wind turbine loading.

Inital studies in literature accounting for the effects of forest canopy were performed over forested flat terrains. Brower (2012) showed that the presence of a forest canopy increases the modeling uncertainty by up to a factor of 5 irrespective of the chosen modeling approach. Finnigan (2000) parameterized the effect of foliage and forest canopy by accounting for the drag force in the momentum equation. The effect of the canopy on turbulence was further taken into account by Sogachev and Panfyorov (2006) and Sogachev (2009) by adding additional source terms for turbulence kinetic energy and turbulence dissipation rate. Desmond et al. (2017) modeled both forest canopy and buoyancy effects by modeling them using source and sink terms and showed that thermal stratification plays an important role in determining the flow over canopies. The canopy model is typically implemented by specifying the tree height and leaf area density (LAD), which represents the area of the leaf and branches of trees. These parameters are typically defined from field measurements or LiDAR point clouds generated from field measurement campaigns and aerial surveys (Queck et al., 2012).

Validation of numerical models for complex terrains requires field measurements. One of the earliest field campaigns was at Askervin hill (Salmon et al., 1988), a smooth isolated hill. Similarly, the Bolund hill campaign was another campaign over an isolated hill in Denmark as detailed by Bechmann et al. (2009), with data under conditions that could be classified as neutral. However, no canopy is seen on these sites. The Alaiz field campaign was performed over a large-scale homogeneous mountain valley topography in Spain with forested regions (Rodrigo et al., 2021). Chavez et al. (2014) showed that using a canopy model improved the accuracy of velocity and turbulence predictions for simulations over Alaiz.

More recently, a field experiment campaign has been carried out over the complex terrain site at Perdigão, Portugal by (Fernando et al., 2018), which is a double ridge with a forested canopy experiencing different thermal stratification conditions. Significant field data is available for further studies, and these are elaborated in Section 2. Several numerical studies have been performed for validation with field data. Laginha Palma et al. (2020) studied the choice of using the appropriate grid size for numerical modeling. Quimbayo et al. (2022) implemented a forest parameterization in the Weather Research and Forecasting model (WRF) with an average tree height of 30 m, resulting in an improvement in the prediction of near-surface wind speeds. This tree height was chosen for practical reasons related to the model setup and is not representative of actual tree height. There is a gap in the existing literature on microscale simulations using a canopy model on a heterogeneous forested complex terrain with a rich dataset such as Perdigão.

## 1.1 Objectives and outline

This paper aims to evaluate the impact of different physical source terms and turbulence models when performing CFD simulations in complex terrain and compare simulation prediction with field measurement profiles at the different groups of towers located at the Perdigão test site. The influence of the following parameters are investigated: Canopy effects, Coriolis force, and the effect of using two different sets of inflow profiles. One idealized set with a log-law velocity profile, and one set of fully developed profiles based on a precursor simulation. Furthermore, the influence of wind direction on the prediction of wind profiles is studied for the different case setups.

An improved understanding of the importance of these phenomena will enable the development of more efficient and reliable tools to perform wind simulations in complex terrain. This paper is structured as follows: a detailed description of the methodology is provided in Section 2, covering the computational domain and meshing, the numerical settings, and different modeling capabilities that have been added as source terms to the conservation equations. The results are presented and discussed in Section 3, in terms of different groups of towers. Section 4 is dedicated to concluding remarks and future perspectives. Finally, additional details on the grid spacing, inflow profiles and wind direction study are provided in the Appendix.

## 2 Perdigão Field measurement campaign

An intensive observation period was carried out from 1 May 2017 to 15 June 2017 at the double ridge site of Perdigão, Portugal, by a consortium of American & European universities (Fernando et al. 2018). The location of the different met masts of heights 60 m and 100 m used in this study is shown in Fig. 2, which could be grouped by their location on top of the Southwest ridge, inside the valley, or at top of the Northeast ridge. From the analysis of the wind rose, the predominant directions at Tower 20 and Tower 29 are perpendicular to the ridges, i.e Northeast and Southwest directions, as shown in Fig. 1, while at Tower 25, flow turning is seen and the dominant wind direction is from the South.

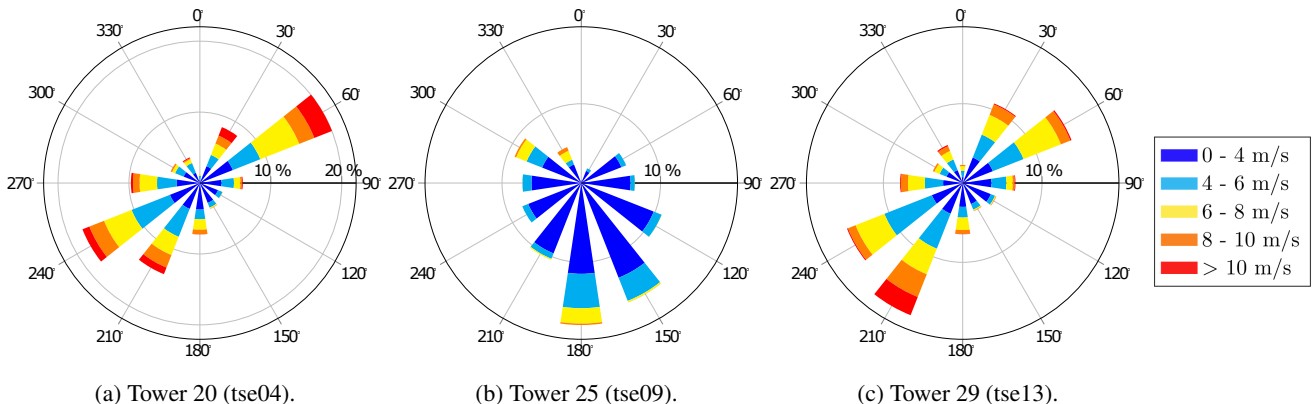

(a) Tower 20 (tse04).     (b) Tower 25 (tse09).     (c) Tower 29 (tse13).

**Figure 1.** Wind rose for 1 month period of May 2017 for towers on top of the ridges and inside the valley.

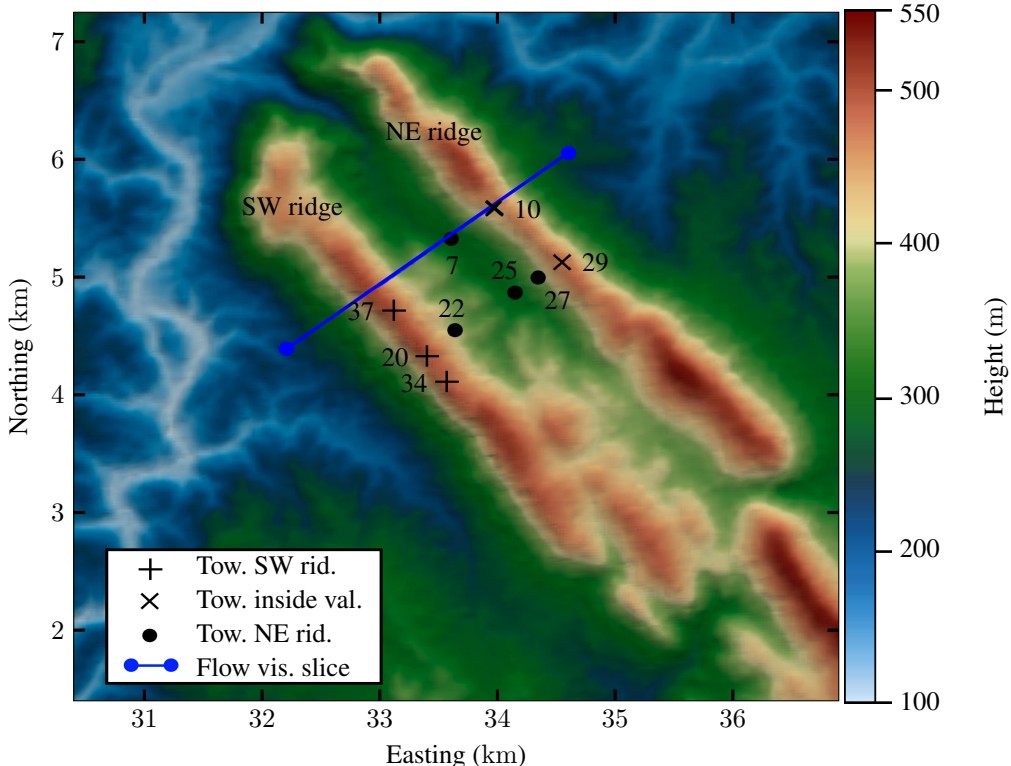

**Figure 2.** Elevation map and locations of interest at Perdigao. Positions of the measurements towers in the SW ridge group are indicated with + symbols, likewise are NE masts marked with x, and the masts in the inside valley group are indicated with black dots. The blue dots and the line represents the flow visualization slice used in Fig. 15. PT-TM06/ETRS89 coordinate system, height above sea level.

A stationary period was found on the date of 4th May, 2017 for the 30-minute averaged time interval of $22:00 - 22:30$ tilt corrected high-frequency dataset from NCAR-EOL, based on the conditions at Tower 20 on top of the ridge, which corresponded to a Bulk Richardson number of approximately -0.03 which qualifies as near-neutral condition. The Bulk Richardson number as defined by Kaimal and Finnigan (1994) is calculated using Eqn. 1. However, the flow conditions are non-neutral at other met mast locations.

$$Ri_b = \frac{g(\bar{\theta}_{100} - \bar{\theta}_2)\Delta z}{\bar{\theta}_{100}[U_{100}^2 + V_{100}^2]} \tag{1}$$

where $\theta$ is the potential temperature, $g$ is the acceleration due to gravity, $U, V$ are the wind velocity components at the reference height. The same period was simulated by Laginha Palma et al. (2020) based on the stationary periods predicted by Carvalho (2019).

## 3  Methodology

The terrain is a $7.5\,\mathrm{km} \times 7.5\,\mathrm{km}$ square centralized around a $100\,\mathrm{m}$ met-mast located on the Southwest ridge. Fig. 3 presents the computational domain and the dimensions are listed on the right side of the figure.

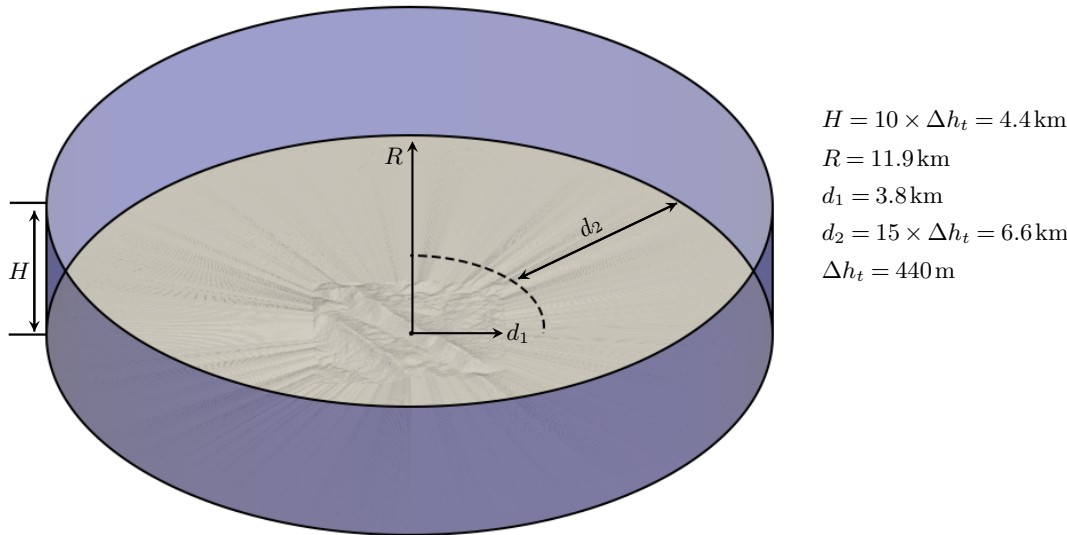

$$H = 10 \times \Delta h_t = 4.4\,\mathrm{km}$$
$$R = 11.9\,\mathrm{km}$$
$$d_1 = 3.8\,\mathrm{km}$$
$$d_2 = 15 \times \Delta h_t = 6.6\,\mathrm{km}$$
$$\Delta h_t = 440\,\mathrm{m}$$

**Figure 3.** Computational domain, $\Delta h_t$ is the difference in the elevation height of the terrain.

A cylindrical computation domain was developed, which provides the flexibility to simulate wind from any wind direction. The authors have successfully applied this approach in previous studies pertaining to urban areas (Hågbo et al. 2021) ( ). A smoothing region from complex to flat terrain was applied towards the outer boundaries, with a minimum radial distance of $15\,\Delta h_t$. Several best practice guidelines have been formulated for grid generation for simulating complex terrain sites, such as by Sørensen et al. (2012) and Laginha Palma et al. (2020) and have been closely followed. The height of the domain is set to ten times the difference in the elevation height of the terrain, $\Delta h_t$, as recommended by Sørensen et al. (2012) when simulating wind flow over complex terrain sites. It consists of approximately 88 million cells ("Fine" case as detailed in the Appendix 5) and is produced with `terrainBlockMesher` developed by (Schmidt et al., 2012), capable of generating structured meshes over complex terrain exclusively consisting of hexahedra cells. The `terrainBlockMesher` tool uses a blending function to smooth the transition from the terrain patch to the outer cylindrical block. Around 50 radial block cells are defined and a radial grading factor is used to enable a stretching in the horizontal direction to cluster cells across the center of the domain and expand towards the boundaries. In terms of the number of cells per main direction ($N_x \times N_y \times N_z$) the mesh comprises of 600 x 600 x 170 across the terrain patch. The vertical mesh resolution is 24.5 m with uniform stretching applied across the entire domain. The minimal mesh height $\Delta z$ next to the ground is close to 3 m. The average value of the wall $y^+$ is around 32,000. The mesh also follows the recommendation for having at least three cells from the ground to the height at the first sampling point for comparison with field measurements. The horizontal mesh resolution over the terrain was set

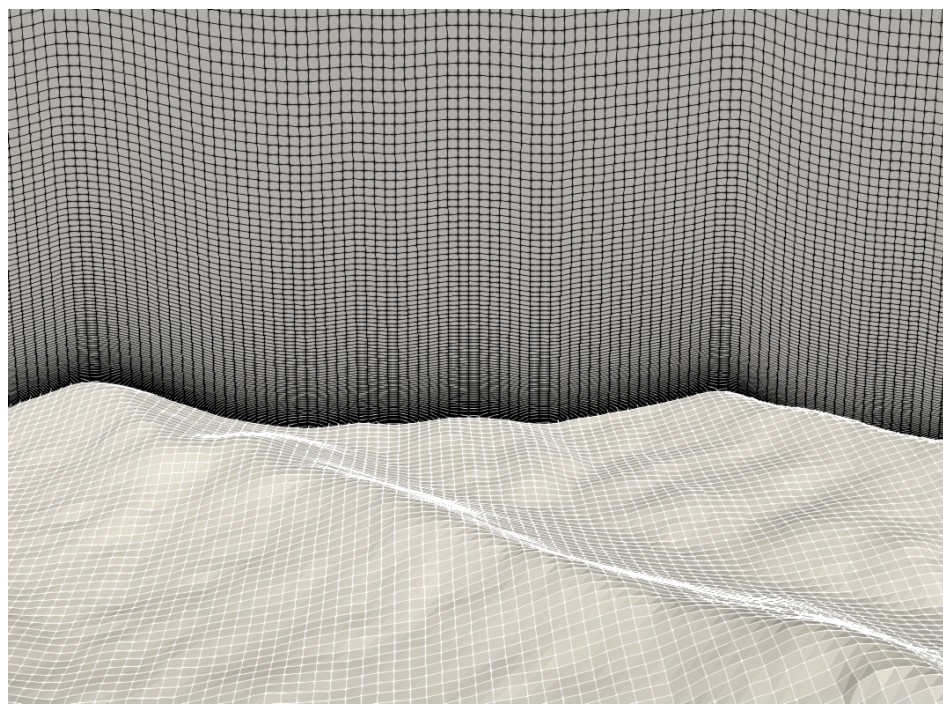

**Figure 4.** Computational grid structure, partial view from the side. The mesh used in this study consists of about 88 million cells, while the mesh presented in this figure is far coarser (about 10 million) for illustration purposes.

close to $12.5\,\mathrm{m}$. It satisfies the minimum resolution of $40\,\mathrm{m}$ recommended specifically for the Perdigão site by Laginha Palma et al. (2020). Fig. 4 illustrates the grid structure used inside the domain from the side. Terrestrial data was obtained from the Shuttle Radar Topography Mission (SRTM) database of the Perdigão field experiment (Fernando et al. 2018). A grid sensitivity
study was performed and presented in Appendix. The results obtained with 5 different grids of increasing mesh resolution show significant differences in the wind profiles especially close to the ground and inside the valley. Hence, a further grid spacing study has to be done as a part of future work and the results presented could change if a finer resolution is used.

All simulations have been conducted using the OpenFOAM (version 2012) toolbox. The simulations are steady-state and performed by solving the incompressible, three-dimensional steady Reynolds-averaged Navier-Stokes (RANS) equations with
the finite volume method. Second-order discretization schemes were used for spatial discretization. The initial iterative convergence criteria were that the scaled residuals should drop four orders of magnitude for all flow variables as per the BPGs. Two steady-state solvers for turbulent flow of incompressible fluids have been used: `buoyantBoussinesqSimpleFoam` (BBSF) and `simpleFoam` (SF). All thermal effects are neglected in the simulations using both the solvers, such that the atmospheric stability of the simulated atmosphere is always neutral. In these simulations, the air density is assumed constant, and
the gravitational force is neglected. The BBSF solver is capable of simulating the effect of buoyancy forces, but these source terms are set to zero for the present neutral case. However, in addition to solving the continuity equation and the momentum

equation, which is solved using SF, the energy equation is also included, allowing for the modeling of non-neutral atmospheric conditions.

## 3.1 Inlet profiles and boundary conditions

Two sets of fixed inlet profiles have been used, both forming a homogeneous atmospheric boundary layer (ABL), either idealized with a logarithmic velocity profile or fully developed profiles obtained from a precursor simulation.

The idealized profiles provide the turbulence quantities and a log-law type ground-normal wind velocity based on the generalization and modification of the well-used set of equations from Richards and Hoaxey (1993). This modification has been implemented by Yang et al. (2009) and uses a more mathematically consistent formulation allowing the turbulent kinetic energy
to vary with height. Also, using a log-law wind velocity profile and the associated turbulent inflow conditions are considered only to be valid in the atmospheric surface layer, which can be roughly estimated as $10\%$ of the atmospheric boundary layer (Temel et al. 2018). The idealized velocity profile has no wind veer, meaning it does not have a changing wind direction with height.

An alternative to using the idealized profiles is using fully developed profiles obtained from a one-dimensional precursor
simulation following the strategies of Koblitz (2013), Alletto et al.(2018). Cyclic conditions are applied on the sides, and the simulations are run for sufficient iterations to allow the development of the profile. The setups are identical to the setups used in the final (successor) simulations except for the computational domain and mesh, the cyclic conditions applied, and the number of iterations. It enables the precursor simulations to produce inlet profiles valid for various conditions, including non-neutral atmospheric conditions. Adjusting the parameters of these source terms is an efficient way to obtain the desired
inflow. The developed velocity profile has wind veer, meaning the wind turns clockwise with height caused by the balance between friction, the Coriolis force, and the pressure gradient force, and that Perdigão is located in the Northern Hemisphere. : Also, the roughness length, $z_0$, was adjusted in the precursor simulations to obtain the right turbulent kinetic energy, TKE, at reference height. The inflow profiles are added in Appendix 5.

An overview of the boundary conditions (BCs) used is presented in Table 1.

**Table 1.** Overview of boundary conditions applied in the neutral simulation setup.

| Patch | Boundary condition type |
|---|---|
| Sides | Inlet/outlet |
| Inlet | Fixed value profiles of an atmospheric boundary layer |
| Outlet | Zero gradient, and fixed pressure or fixed $(p - (\rho g h))$ |
| Terrain | Rough wall, $z_0 = 0.02\,\mathrm{m}$ |
| Top | Slip |

Robin boundary conditions that act as an inlet or outlet are used on the sides of the domain to simulate wind from any horizontal direction. The direction of the flux automatically determines the inlet and outlet regions. The inlet profiles are fixed to form a homogeneous atmospheric boundary layer (ABL), either idealized profiles representing neutral atmospheric

conditions or fully developed profiles obtained from precursor simulation based on the work of (van der Laan et al., 2020). Outlet conditions were used with constant pressure in the simulations using the SF solver, or equally, constant $(p-(\rho g h))$ in the simulations using the BBSF solver. Zero gradient was set for the remaining variables. The direction of the flux automatically determines the inlet and outlet regions. No-slip conditions with wall functions were used for the terrain. Specifically, a rough wall condition was applied (Hargreaves and Wright, 2007). The roughness length was set to $z_0 = 0.02\,\mathrm{m}$, based on average values of a roughness length map provided in the database of the Perdigão field experiment (Fernando et al., 2018).

## 3.2 Source terms

Three different source terms have been added to the momentum equation for some simulation cases: Tree canopy as a porous medium, Coriolis force, and pressure gradient force. Additionally, a limiter in the turbulence dissipation equation was added to one of the cases.

The canopy is modeled as a porous medium based on the work of Costa (2007). A drag term is added to the momentum equation. The canopy source term was utilized with the `simpleFoam` (SF) flow solver. The porosity model is based on the `powerLawLopesdaCosta` model implemented in OpenFOAM. It is a variant of the power law porosity model with a spatially varying drag coefficient. This source term is applied to the momentum equation to reproduce the momentum dissipation that the trees and their foliage should produce in the flow. The following parameters are used: Porosity surface area per unit volume or the leaf area density ($\Sigma = 1.0$), Drag coefficient ($C_d = 0.25$), and the Power law model exponent coefficient ($C1 = 2.0$) for the Equation 2. A mean tree height of $3\,\mathrm{m}$ is chosen based on the mean canopy height at different locations provided by Vasiljevic et al. (2017). This was applied all over the entire flow domain. A cell set was used to select a volume of cells 3 m above the ground (which forms a canopy zone where the source terms are activated). An overall tree height of $3\,\mathrm{m}$ is also supported by analyzing a LiDAR point cloud acquired from the database of the Perdigão field experiment (Fernando et al., 2018) as shown in Fig. 5. The source term in the canopy model for velocity is as shown below:

$$S_p -= \alpha \rho \left( C_d \Sigma \left| u_o \right| \right) u \tag{2}$$

where $\Sigma$ is the Leaf area density, $C_d$ is the plant canopy drag coefficient. As highlighted in Lalic and Mihailovic (Lalic and Mihailovic, 2004), forests have a higher foliage density at the half top than at the bottom zones. However, for the present case, simulations are performed with a uniform leaf area density.

The Coriolis force source term is included as a momentum source term ($U$ and $V$ momentum equations). The Coriolis force is calculated based on the planetary rotational period ($\Omega = 24\ h$) and the latitude ($\lambda$) for Perdigão (39.68 °N), where $f_c = 2\Omega\sin(\lambda)$. Similar to Koblitz (2013), the Coriolis force in the vertical direction is neglected since it is negligible compared to the gravitational acceleration.

A pressure gradient forcing term is also included in the simulation cases using the BBSF solver. In setups using this solver, the wind is driven by the balance of the lateral pressure gradient force and the Coriolis force, which are added in the momentum

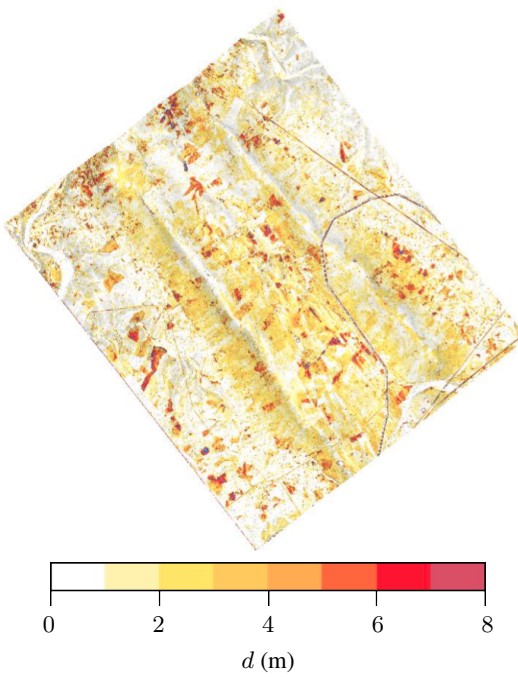

**Figure 5.** Minimum Euclidean distance, $d$, from the trees/vegetation to the bare ground effectively showing their height. Results obtained from analyzing a LiDAR point cloud acquired from the database of the Perdigao Field Experiment website *https://perdigao.fe.up.pt/*.

equation as shown in equations provided in the Appendix 5. This balance is called geostrophic balance, and the resulting wind is a geostrophic wind, which in the Northern Hemisphere goes counterclockwise around low-pressure systems.

The level of turbulent kinetic energy is controlled in the domain by the use of a parameter called the maximum turbulent mixing length scale, used to limit the production and dissipation terms of the turbulence model. Ambient source terms were added to avoid zero turbulence values above the ABL similar to those mentioned in (van der Laan et al., 2021a) and applied to the entire domain. The values were set to $k_{Amb} = 0.001$ and $\epsilon_{Amb} = 7.208e - 08$ as summarized in Table 3. The geostrophic wind is denoted by G and the Coriolis frequency parameter is denoted by $f_c$. The governing equations for the BBSF solver and the different source terms are further detailed in Appendix 5.

### 3.3 Simulation cases

The setup for all the simulations was identical except for certain parameters. The solver used is either `simpleFoam` (SF) or `buoyantBoussinesqSimpleFoam` (BBSF). The inlet profiles, either idealized with a logarithmic velocity profile representing neutral atmospheric conditions or fully developed profiles obtained from precursor simulation. Source terms added, including canopy (tree height set to $3\,\mathrm{m}$) and the Coriolis force. Three turbulence models have been used: the standard k-$\epsilon$ turbulence model (SKE), a modified k-$\epsilon$ turbulence model (MKE), and the k-$\omega$ turbulence model (KO). The MKE model is applied in the simulation case where trees/vegetation (canopy) has been added as a porous medium with the SF solver. The

turbulent constants are modified with $C_\mu = 0.033$. This model was tuned using experimental data and LES (Large Eddy Simulation) for a homogeneous forest (Costa 2007). The boundary conditions are overall the same but had to be adjusted to account for the different solvers and inlet profiles used. An overview of all the simulation cases is provided in Table 2. The set of chosen turbulence constants is shown in Table 3.

**Table 2.** List of simulation cases simulating a period of neutral atmosphere, 22:00-22:30 (04.05.17).

| Case name | Solver | Inlet profiles | Source terms | | | Turbulence mod. |
|---|---|---|---|---|---|---|
| | | | Canopy | Coriolis | Pres. gr. | |
| SF1 | SF | Idealized | No | No | No | SKE |
| SF2 | SF | Idealized | No | No | No | KO |
| SF3 | SF | Idealized | Yes | No | No | MKE |
| SF4 | SF | Idealized | No | Yes | No | SKE |
| BBSF1 | BBSF | Precursor | No | Yes | Yes | KE-Lim |

Numerical convergence difficulties can be encountered when using the global turbulence length scale limiter (van der Laan et al., 2021b) when applied to the present case for simulations over the complex terrain of Perdigão. These convergence difficulties can be particularly challenging when setting the values of the maximum limiting length scale to low values. Here, the model is trying to restrict the turbulence scale, but physically the large length scales of turbulence are produced from hills or features that are in the order of the maximum value that is set by turbulence model.

## 4 Results

The results are discussed based on three groups of towers of interest: on top of the Southwest ridge, inside the valley and on top of the Northeast ridge. The inlet profiles for all the simulations are calibrated to match the measured velocity magnitude and direction at 100 m at tower 20, which corresponds to an elevation of 573 m. Different metrics are analyzed for all the simulation models at different towers and are further explained. The root mean square error (RMSE) between the averaged profiles of the measured data and the simulated results is computed and presented in Table 4. A hit rate metric for a given model is defined as the number of predictions within one standard deviation over each height of the field measurement. The relative error for the turbulent kinetic energy and wind direction is shown in Tables 5, 6 respectively. The relative error is given in percentage and is the difference between the simulated and field measurement divided by the field measurement value. The error bars represent the standard deviation of the mean measurements.

### 4.1 Model prediction on SW ridge

The towers on the SW ridge as shown in Fig. 2, are the Tower 20, 34 and 37. This is a region of flow acceleration. The inflow profile is calibrated to match the velocity at 100 m height for Tower 20 as shown in the wind velocity profiles shown in Fig. 6 (a)

**Table 3.** Turbulence constants for different turbulence models.

| Coefficient | Turbulence model | | | |
|---|---|---|---|---|
| | SKE | MKE | KO | KE-Lim |
| $C_\mu$ | 0.09 | 0.033 | 0.09 | 0.09 |
| C1 | 1.44 | 1.44 | - | 1.44 |
| C2 | 1.92 | 1.92 | - | 1.92 |
| $\sigma_\epsilon$ | 1.30 | 1.85 | - | 1.30 |
| $\sigma_K$ | 1.0 | - | 0.5 | - |
| $\alpha_K$ | - | | 0.5 | - |
| $\alpha_\omega$ | - | - | 0.6 | - |
| $\beta*$ | 0.09 | - | - | 0.09 |
| $\beta$ | - | - | 0.072 | - |
| $\nu$ | 1.5e-05 | 1.5e-05 | 1.5e-05 | 1.5e-05 |
| $L_{max}$ | - | - | - | 62.14 |
| $k_{Amb}$ | - | - | - | 0.001 |
| $\epsilon_{Amb}$ | - | - | - | $7.208e-08$ |
| $T_{Ref}$ | - | - | - | 300 |
| $Pr$ | - | - | - | 0.9 |
| $Prt$ | - | - | - | 0.74 |
| $G$ | - | - | - | 6.2 |
| $f_c$ | - | - | - | 0.929e-4 |

at 100 m. All models except the canopy model (SF3) show an over prediction of the velocity profiles close to the ground at Tower 20. The canopy model (SF3) underpredicts the velocity close to the ground but accurately captures the shape of the profile. This prediction could be improved by considering a non-uniform tree height and removing the canopy source terms on top of the ridges. On the other hand, at Tower 34, shown in Fig 6 (b), the canopy model (SF3) shows a good match with the field measurement, and on Tower 37 seen in Fig 6 (c), shows a slight under prediction. In models that do not account for a canopy (SF1, SF2, SF4 and BBSF1), a speed-up is seen close to the ground with large RMSE errors as shown in Table 4. The results suggest that the inclusion of the canopy is a good choice for predicting of velocity profiles. However, the canopy parameters need to be adapted based on the forest point cloud data. BBSF1 model (k-$\epsilon$ Lim) shows a higher speed-up close to the ground at most of the towers. The turbulent kinetic energy profiles are shown in Fig 7, showing a good prediction for all the models except the Canopy model (S3) which produces excessive turbulence levels compared to field measurements. Fig 8 shows the predicted wind direction profiles at the three towers showing a good prediction for most of the models within one standard deviation of the measurements. As highlighted in the methodology section, there is a large uncertainty in wind direction seen in the field experiment that varies with location and height which is further investigated in Section 4.4 for two additional wind directions. This uncertainty highlights a limitation in the models' ability to represent actual conditions for such a complex

**Table 4.** RMSE for wind speed predictions on the main towers of interest.

| Case | Weather mast and group | RMSE (m/s) Height over terrain | | | | | | | Average RMSE (m/s) | Hit rate (max 7) |
|---|---|---|---|---|---|---|---|---|---|---|
| | | 10 m | 20 m | 30 m | 40 m | 60 m | 80 m | 100 m | | |
| SF1 (k-ε) | Tower 20 (SW rid.) | 0.70 | 0.16 | 0.14 | 0.11 | 0.17 | 0.05 | 0.05 | 0.20 | 6 |
| | Tower 25 (valley) | 1.40 | 1.3 | 1.21 | 1.01 | 0.73 | 0.49 | 0.27 | 0.92 | 6 |
| | Tower 29 (NE rid.) | 1.81 | 1.18 | 1.01 | 0.67 | 0.43 | 0.58 | 0.71 | 0.91 | 3 |
| SF2 (k-ω) | Tower 20 (SW rid.) | 0.73 | 0.21 | 0.19 | 0.16 | 0.18 | 0.04 | 0.05 | 0.22 | 6 |
| | Tower 25 (valley) | 1.89 | 1.91 | 1.93 | 1.9 | 1.83 | 1.68 | 1.51 | 1.81 | 0 |
| | Tower 29 (NE rid.) | 1.59 | 0.86 | 0.71 | 0.43 | 0.22 | 0.43 | 0.61 | 0.69 | 6 |
| SF3 (canopy) | Tower 20 (SW rid.) | 0.65 | 0.53 | 0.46 | 0.38 | 0.37 | 0.19 | 0.06 | 0.38 | 2 |
| | Tower 25 (valley) | 0.37 | 0.42 | 0.36 | 0.21 | 0.1 | 0.37 | 0.57 | 0.34 | 1 |
| | Tower 29 (NE rid.) | 2.03 | 2.56 | 2.77 | 2.7 | 2.75 | 2.73 | 2.71 | 2.61 | 0 |
| SF4 (Coriolis) | Tower 20 (SW rid.) | 0.75 | 0.31 | 0.28 | 0.24 | 0.29 | 0.15 | 0.14 | 0.31 | 6 |
| | Tower 25 (valley) | 0.88 | 1.24 | 1.3 | 1.22 | 1.25 | 1.21 | 1.29 | 1.20 | 5 |
| | Tower 29 (NE rid.) | 1.47 | 0.59 | 0.29 | 0.09 | 0.47 | 0.29 | 0.24 | 0.49 | 6 |
| BBSF1 (k-ε Lim) | Tower 20 (SW rid.) | 1.33 | 0.73 | 0.71 | 0.66 | 0.37 | 0.41 | 0.35 | 0.65 | 4 |
| | Tower 25 (valley) | 1.15 | 1.46 | 1.53 | 1.51 | 1.61 | 1.64 | 1.7 | 1.51 | 3 |
| | Tower 29 (NE rid.) | 1.41 | 0.66 | 0.5 | 0.24 | 0.2 | 0.35 | 0.56 | 0.56 | 6 |

site. In terms of the turbulence model, the SF1 model (k-ε) is seen to produce a slightly greater speed-up compared to the SF2 model (k-ω). Still, overall the prediction capability for the wind speed, turbulent kinetic energy, and wind speed is similar.

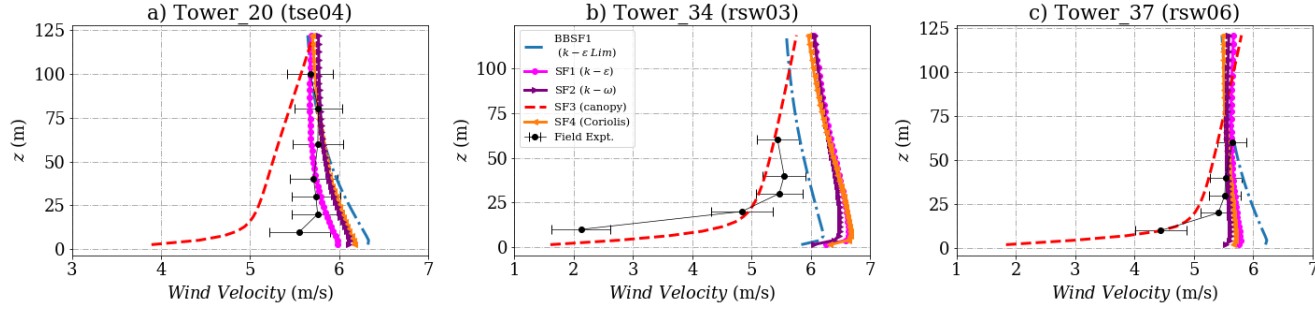

**Figure 6.** Simulation results and experimental data for wind velocity on the Southwest ridge for a) Tower 20 (tse04) b) Tower 34 (rsw03) c) Tower 37 (rsw06). The locations of the masts are given in Fig 2.

**Table 5.** Relative error and hit rate metrics for turbulent kinetic energy predictions at the main towers of interest.

| Case | Weather mast and group | Average relative difference in TKE (%) | Hit rate (max 7) |
|---|---|---|---|
| SF1 (k-$\epsilon$) | Tower 20 (SW rid.) | 11.61 | 7 |
| | Tower 25 (valley) | 51.24 | 7 |
| | Tower 29 (NE rid.) | 35.53 | 4 |
| SF2 (k-$\omega$) | Tower 20 (SW rid.) | 10.36 | 7 |
| | Tower 25 (valley) | 23.08 | 7 |
| | Tower 29 (NE rid.) | 33.69 | 7 |
| SF3 (canopy) | Tower 20 (SW rid.) | 96.98 | 0 |
| | Tower 25 (valley) | 46.2 | 7 |
| | Tower 29 (NE rid.) | 25.49 | 7 |
| SF4 (Coriolis) | Tower 20 (SW rid.) | 12.61 | 7 |
| | Tower 25 (valley) | 49.91 | 7 |
| | Tower 29 (NE rid.) | 55.51 | 2 |
| BBSF1 (k-$\epsilon$ Lim) | Tower 20 (SW rid.) | 10.21 | 7 |
| | Tower 25 (valley) | 51.57 | 7 |
| | Tower 29 (NE rid.) | 55.44 | 1 |

**Table 6.** Relative error and hit rate metrics for wind direction predictions at the main towers of interest.

| Case | Weather mast and group | Average difference in wind direction ($^\circ$) | Hit rate (max 7) |
|---|---|---|---|
| SF1 (k-$\epsilon$) | Tower 20 (SW rid.) | 1.14 | 6 |
| | Tower 25 (valley) | 14.02 | 7 |
| | Tower 29 (NE rid.) | 8.42 | 4 |
| SF2 (k-$\omega$) | Tower 20 (SW rid.) | 1.18 | 6 |
| | Tower 25 (valley) | 18.39 | 7 |
| | Tower 29 (NE rid.) | 6.45 | 5 |
| SF3 (canopy) | Tower 20 (SW rid.) | 1.3 | 7 |
| | Tower 25 (valley) | 10.93 | 7 |
| | Tower 29 (NE rid.) | 10.25 | 3 |
| SF4 (Coriolis) | Tower 20 (SW rid.) | 1.79 | 6 |
| | Tower 25 (valley) | 49.41 | 2 |
| | Tower 29 (NE rid.) | 3.18 | 7 |
| BBSF1 (k-$\epsilon$ Lim) | Tower 20 (SW rid.) | 5.63 | 1 |
| | Tower 25 (valley) | 41.8 | 3 |
| | Tower 29 (NE rid.) | 4.39 | 7 |

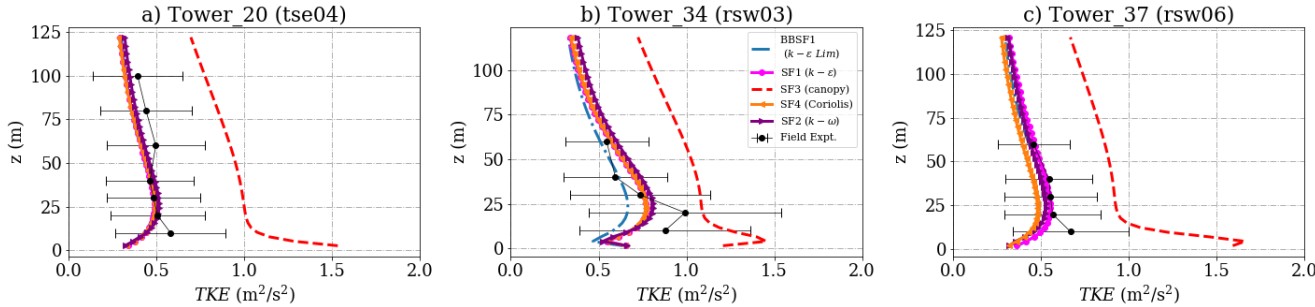

**Figure 7.** Simulation results and experimental data for turbulent kinetic energy on the Southwest ridge for a) Tower 20 (tse04) b) Tower 34 (rsw03) c) Tower 37 (rsw06). The locations of the masts are given in Fig 2.

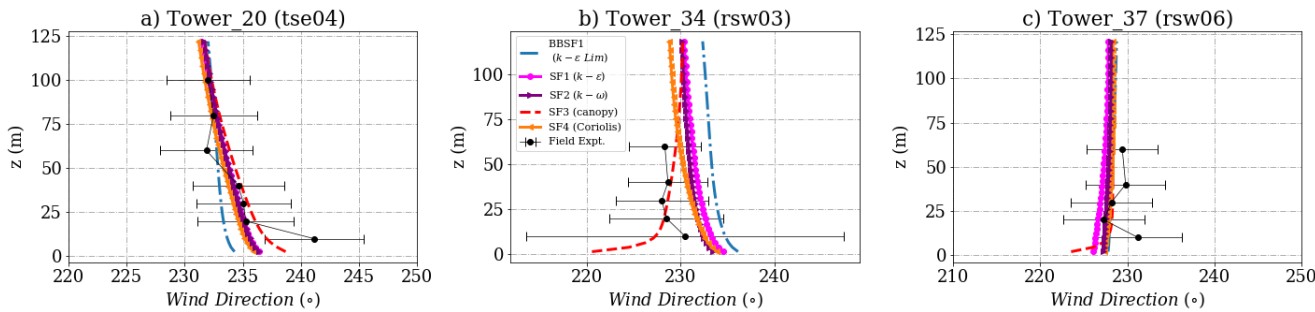

**Figure 8.** Simulation results and experimental data for wind direction on the Southwest ridge for a) Tower 20 (tse04) b) Tower 34 (rsw03) c) Tower 37 (rsw06). The locations of the masts are given in Fig 2.

## 4.2 Model prediction inside the valley

The towers of interest inside the valley are Towers 25, 7, 27 and 22 as seen earlier in Fig 2. The valley is a region of strong
flow separation and flow re-circulation. The predicted velocity profiles are shown in Figs. 9 and 10 with different predictions of
the extent of re-circulation zones, resulting in various shapes of the wind profiles. At Towers 25, 22 and 27 the canopy model
significantly underpredicts the velocity profiles. The SF2 (Canopy), SF1 $(k - \epsilon)$ and SF2 $(k - \omega)$ show the best prediction at
Tower 7, which is located at the lowest altitude among the other towers. Overall the SF1 $(k - \epsilon)$ model shows the best prediction
at all the towers inside the valley.

Figs. 11 and 12 show a comparison of turbulent kinetic energy profiles between the present model predictions and the field
measurements. The measured values of turbulent kinetic energy are high inside the valley at all met-mast locations, indicative
of flow mixing and high turbulence. However, all the models seem to strongly underpredict the turbulent kinetic energy with
large relative errors as seen in Table 5, except for Tower 22, which is located the closest to the Southwest ridge.

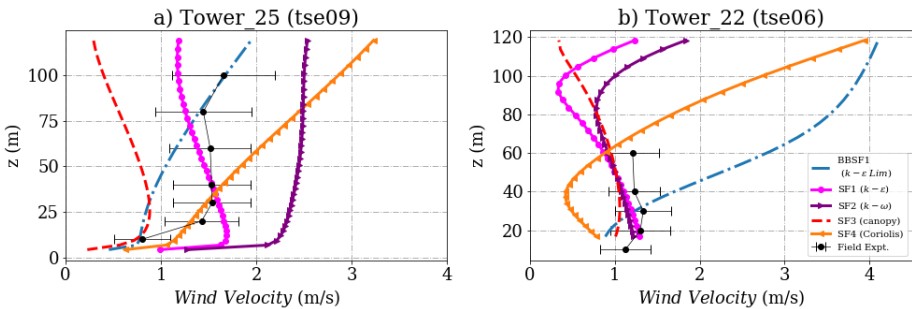

**Figure 9.** Simulation results and experimental data wind velocity inside the valley for a) Tower 25 (tse09) b) Tower 22 (tse06). The locations of the masts are given in Fig. 2.

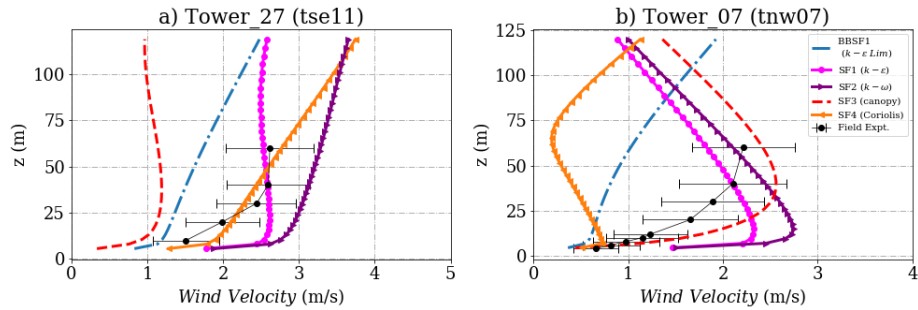

**Figure 10.** Simulation results and experimental data for wind velocity inside the valley for a) Tower 27 (tse11) b) Tower 7 (tnw07). The locations of the masts are given in Fig. 2.

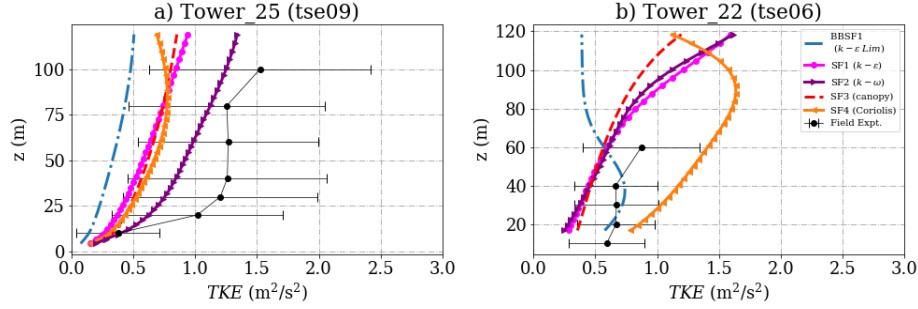

**Figure 11.** Simulation results and experimental data for turbulent kinetic energy inside the valley for a) Tower 25 (tse09) b) Tower 22 (tse06). The locations of the masts are given in Fig. 2.

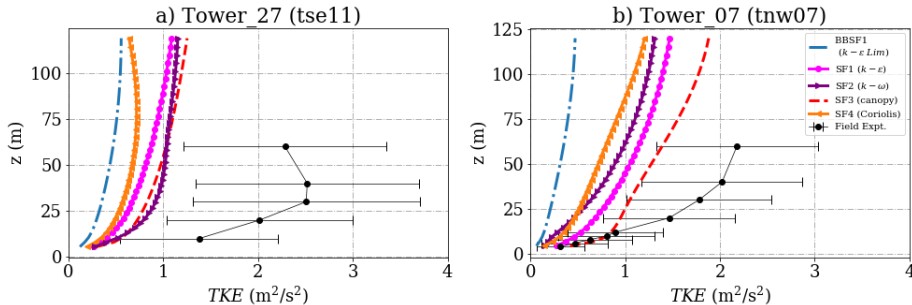

**Figure 12.** Simulation results and experimental data for turbulent kinetic energy inside the valley for a) Tower 27 (tse11) b) Tower 7 (tnw07). The locations of the masts are given in Fig. 2.

The wind direction profiles are shown in Figs. 13 and 14 respectively. A sudden shift in wind direction, indicative of flow separation and re-circulation is seen in Fig. 13. Table 6 provides the hit rate for different models at all the towers. Fig. 15 shows the velocity contour for the predicted re-circulation zone inside the valley for the SF3 model (canopy) and the BBSF1 model (k-$\epsilon$ Lim). The SF3 model (canopy) produces a single large re-circulation zone compared to the other models, and a smaller and double re-circulation zone is seen for the BBSF1 model (k-$\epsilon$ Lim). The large recirculation zone results in a significant under prediction of the velocity profiles seen earlier in Figs. 9 and 10 for the canopy model. Menke et al. (2019) also investigated the flow re-circulation zones around the Perdigão valley under different atmospheric stability conditions.

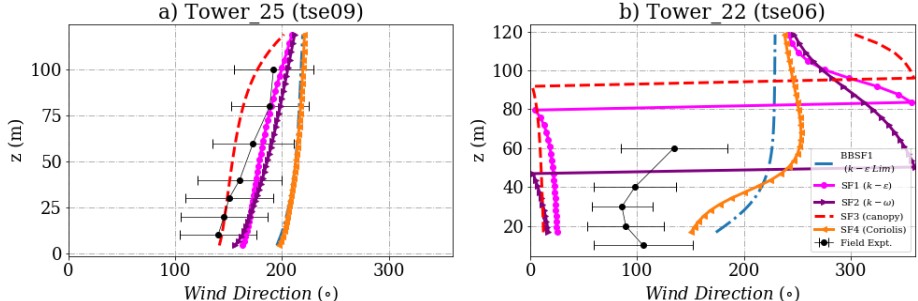

**Figure 13.** Simulation results and experimental data for wind direction inside the valley for a) Tower 25 (tse09) b) Tower 22 (tse06). The locations of the masts are given in Fig. 2.

## 4.3 Model Prediction on NE ridge

The wind velocity profiles for Tower 29 and Tower 10 on the Northeast ridge are shown in Fig. 16. This area on the top of a ridge is a region of flow acceleration. It is just downstream from the re-circulation zone inside the valley, which makes the predictions quite challenging. The SF3 model (canopy) is seen to under predict the velocity profiles significantly on top of this

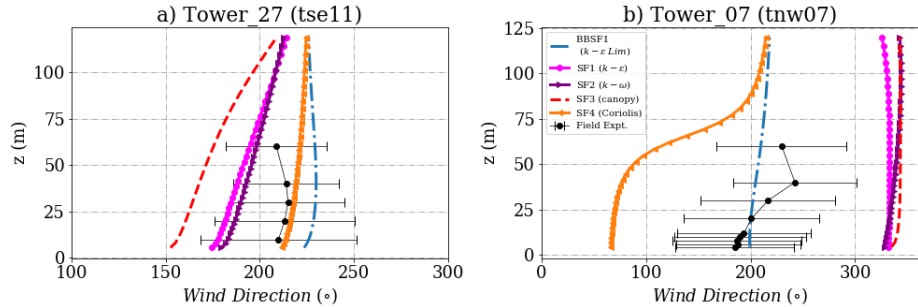

**Figure 14.** Simulation results and experimental data for wind direction inside the valley for a) Tower 27 (tse11) b) Tower 7 (tnw07). The locations of the masts are given in Fig. 2.

ridge. All other models appear to provide a good prediction of the velocity profile around one standard deviation of the field measurements. The region near Tower 10 comprises a mountain gap (Vassallo et al. 2020), where the local flow features could play an important role. The SF3 (Coriolis) and the BBSF1 models predict the profiles close to the field measurement. Excessive turbulence levels are seen close to the ground in the field measurements shown in Fig. 17. All models appear to underpredict the field measurements of turbulence close to the ground, and the SF3 (canopy) shows the closest match. The BBSF1 model (k-$\epsilon$ Lim) underpredicts the turbulence levels due to the use of the maximum length scale limiter. The wind direction profiles at the towers are shown in Fig. 18. As seen for the TKE profiles, a large uncertainty is seen in the wind direction on this ridge, this is expected as the wind is from the Southwest direction. Most of the model predictions fall within one standard deviation of the measurements at Tower 29. As seen in Fig. 15, the prediction of wind direction at Tower 10 is dependent on the extent of re-circulation zone and flow reattachment location on top of the ridge.

## 4.4 Influence of wind direction

The inflow direction plays an essential role in the wind predictions for complex terrains. A wind direction standard deviation of around 7° is seen in the field measurements at Tower 20 (tse04). Simulations have been performed for two additional wind directions $(231 \pm 3.5°)$ for each model to look at the differences observed between the three incoming wind directions.

The results for the model predictions using the SF1 $(k - \epsilon)$ model are presented in this section, while the results obtained using the other models of this study are shown in Appendix. For each model, the wind profiles at three different met-masts, corresponding to Tower 20 (tse04) on the South-West ridge, Tower 25 (tse09) inside the valley, and Tower 29 (tse13) on the North-East ridge are shown.

All simulations are calibrated for Tower 20 as seen in Fig 19 with the same inflow velocity at the reference height of 100 m on the tower and only changing the wind direction. The predicted profiles inside the valley and on the North-East ridge appear to vary quite significantly with the inflow wind direction as seen in Figs 20 and 21. Streamline trajectories for the inflow with three different wind directions at the Towers are shown in Fig 22. Wind passing through Tower 29 with inlet wind from 234.5°

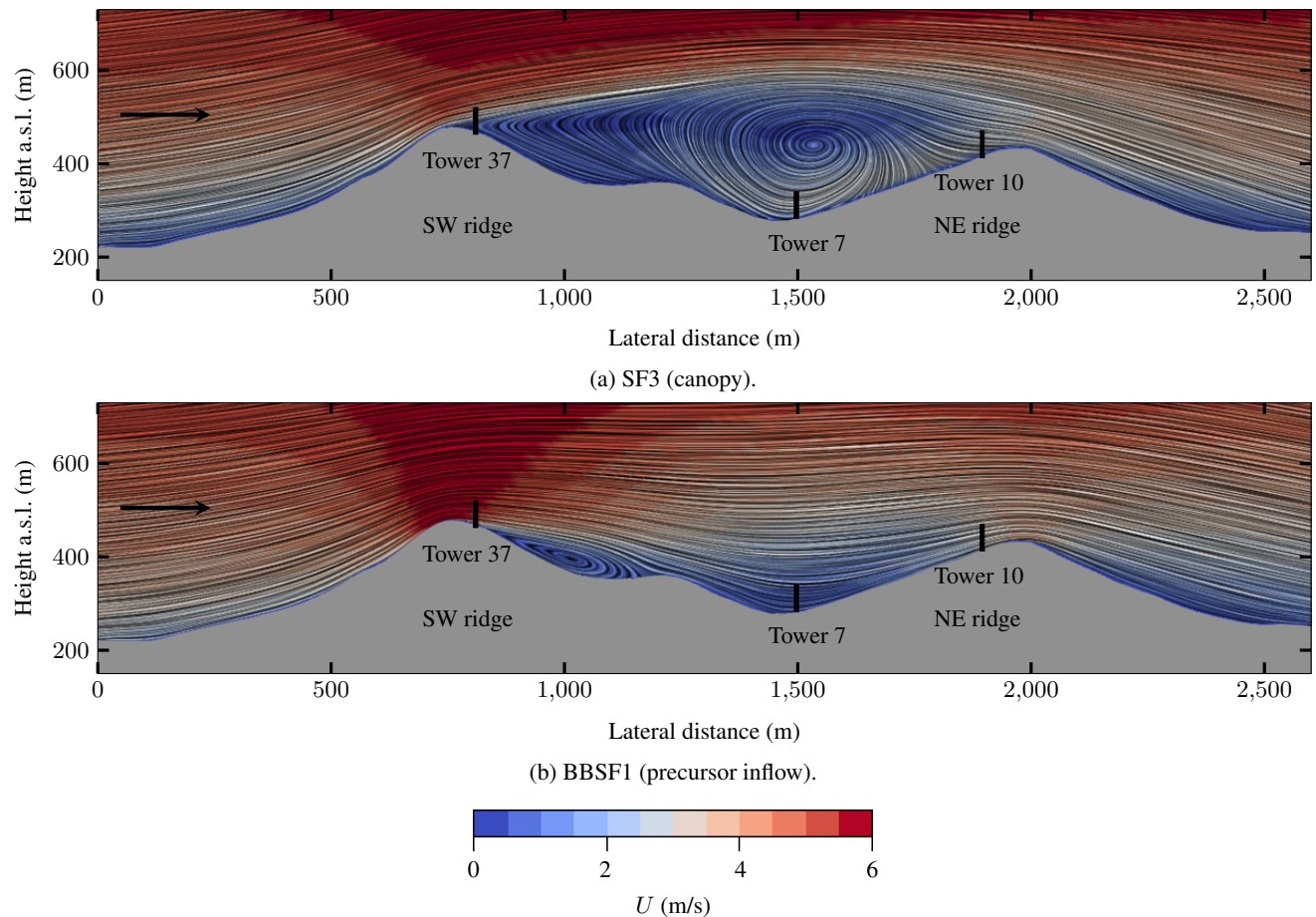

**Figure 15.** Vertical slice illustrating the re-circulation zones inside the valley. Flow patterns and velocity magnitude, $U$, are visualized through the technique line integral convolution (LIC). The black lines represent the location and height of the nearest measurement towers in front of or behind the slice. The black arrows indicate the wind direction at the inlet, wind from SW at $231°$, which is parallel to the orientation of the presented slice. The slice location and the locations of the masts are given in Fig. 2.

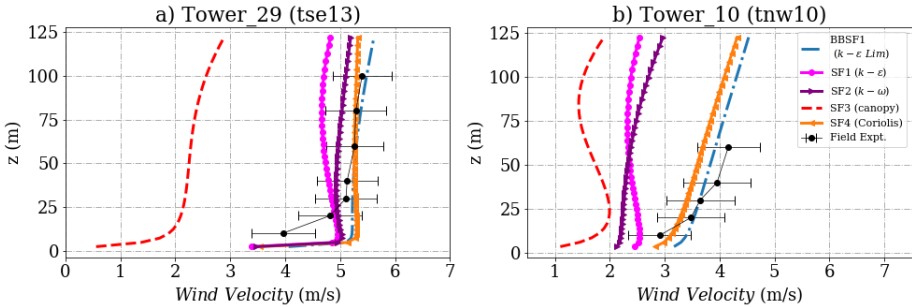

**Figure 16.** Simulation results and experimental data for wind velocity on the Northeast ridge for a) Tower 29 (tse13) b) Tower 10 (tnw10). The locations of the masts are given in Fig. 2.

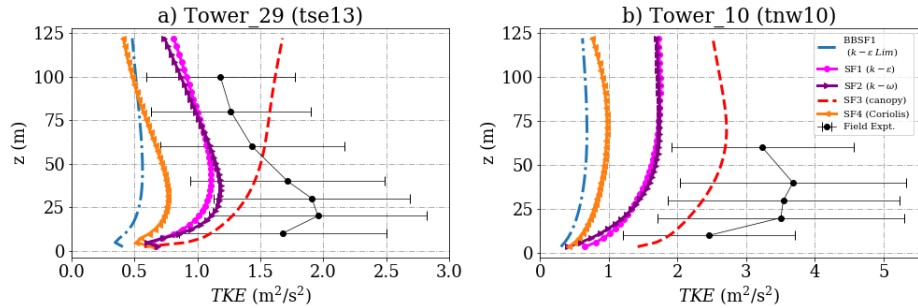

**Figure 17.** Simulation results and experimental data for turbulent kinetic energy on the Northeast ridge for a) Tower 29 (tse13) b) Tower 10 (tnw10). The locations of the masts are given in Fig. 2.

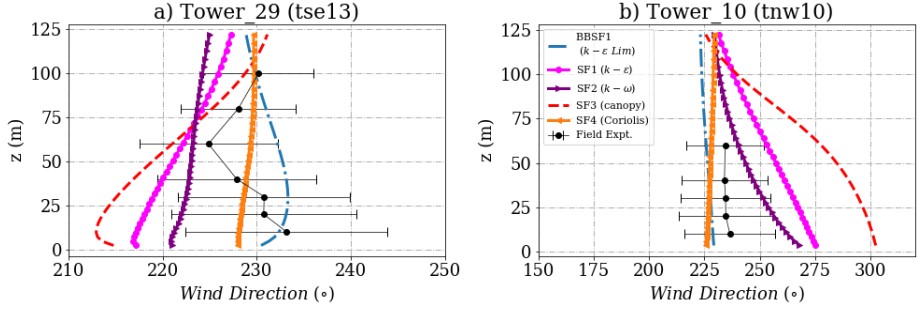

**Figure 18.** Simulation results and experimental data for wind direction on the Northeast ridge for a) Tower 29 (tse13) b) Tower 10 (tnw10). The locations of the masts are given in Fig. 2.

exhibits a very different trajectory compared to the other inlet wind directions and the set of trajectories passing through the mast upstream, Tower 20. The wind here was initially deflected off from the NE ridge and led into the valley by a channelling effect. As a result, the wind speed decreases significantly before passing Tower 29, before re-gaining in intensity downstream

of the ridge as the wind accelerates downhill.

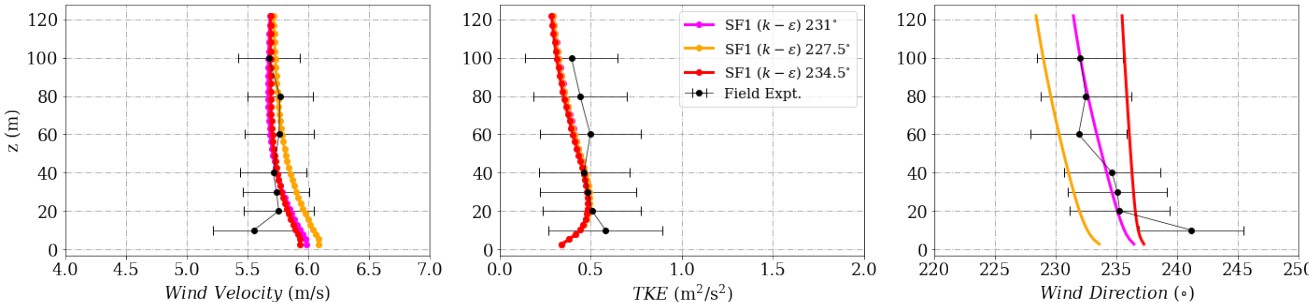

**Figure 19.** Simulation results and experimental data for wind velocity on the Southwest ridge for a) Tower 20 (tse04). The locations of the masts are given in Fig 2.

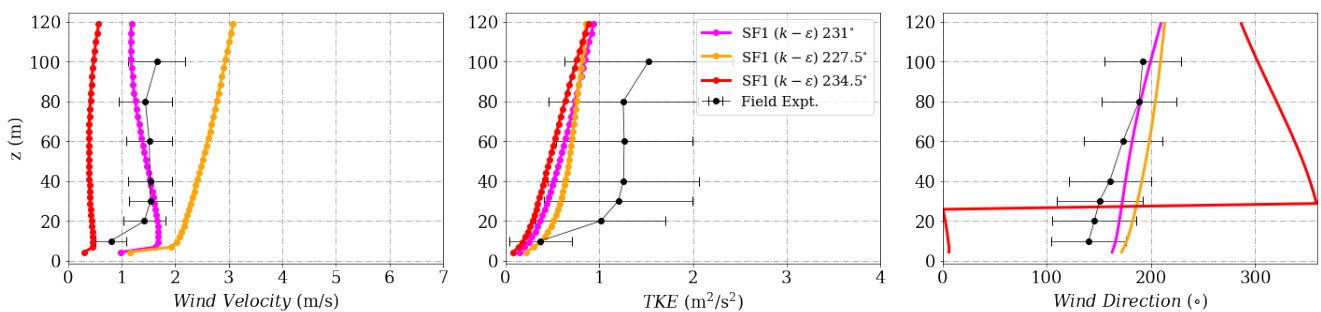

**Figure 20.** Simulation results and experimental data wind velocity inside the valley Tower 25 (tse09).

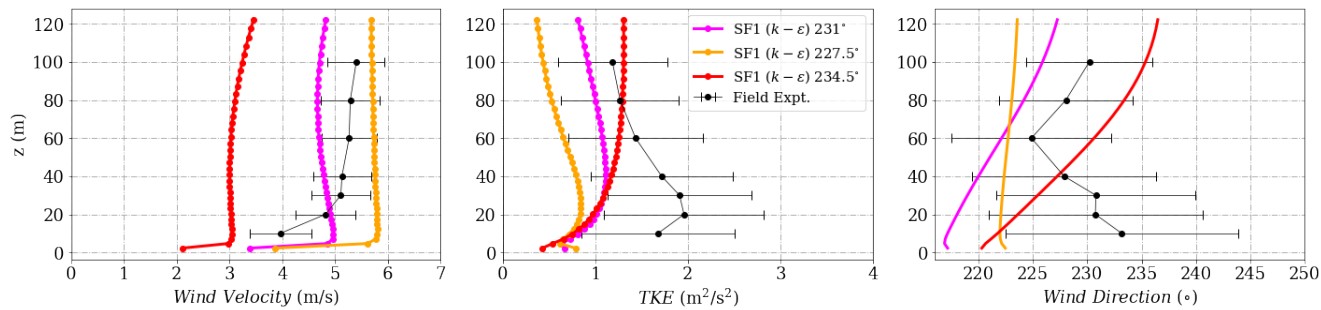

**Figure 21.** Simulation results and experimental data for wind velocity on the Northeast ridge for Tower 29 (tse13).

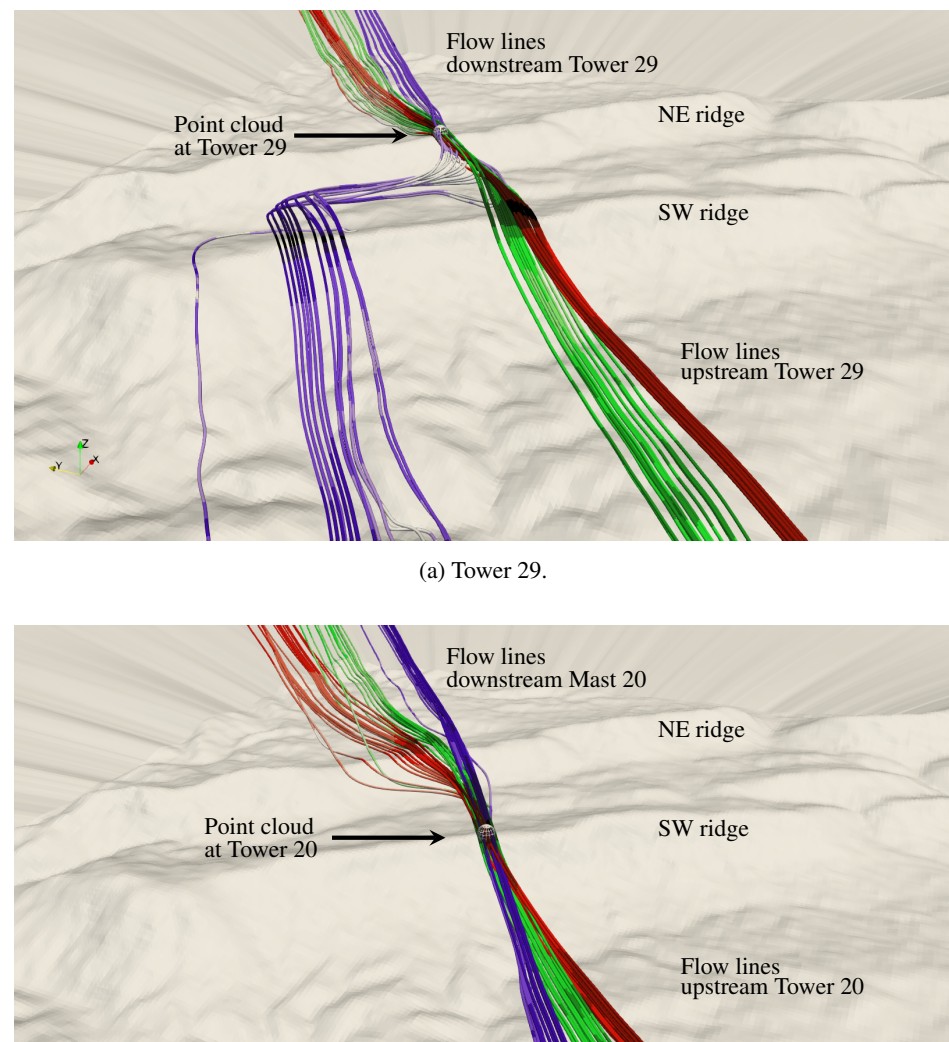

(a) Tower 29.

(b) Tower 20.

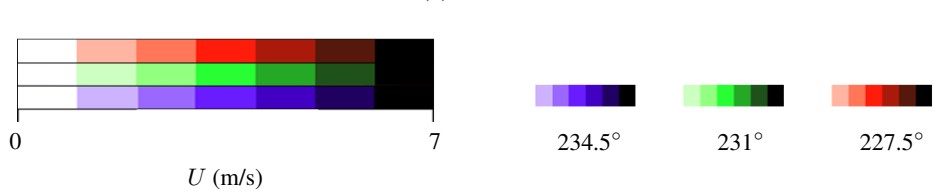

$U$ (m/s)

234.5°        231°        227.5°

**Figure 22.** Wind paths of air parcels passing through a sphere of 55 m radius placed on top of the ground at given met mast locations. Flow lines colored in green represent trajectories for wind coming from 231° at the inlet, while red and blue lines illustrate wind at the inlet from 227.5° and 234.5° correspondingly.

## 5 Conclusions

A Reynolds-averaged Navier-Stokes (RANS) model is set up using OpenFOAM (version 2012) to simulate a 30-minute averaged stationary period corresponding to near-neutral conditions at met-mast Tower 20 located at the Southwest ridge. In that period and for that tower, the wind comes from Southwest at 231° at 100 m height above ground. Five different models are simulated comprising of different source terms to account for the effects of the canopy, the Coriolis force and pressure gradient force, and two different inflow profiles. One idealized set with a log-law velocity profile, and one set of fully developed profiles based on a precursor simulation. Based on the flow topology, the predicted profiles are analyzed in terms of the different groups of towers on top of the ridges and inside the valley. The complex terrain site of Perdigão represents a large spatial variability of forest canopy and surface elevation, which contribute to variable flow topology at different met-masts. The key conclusions for different groups of towers are summarised as follows:

a) **For the towers on the Southwest ridge:** The region at the Southwest ridge is a zone of flow acceleration at the first oncoming ridge downstream of the inlet for wind coming from the Southwest. The inflow profiles are calibrated to closely match the wind speed and direction at Tower 20. Using a canopy model (SF3) decreases the velocities near the surface and is a closer match with field data at Tower 34 and 37. Other models over-predict the velocity profile close to the ground. However, the canopy parameters need to be tuned as the surface heterogeneity is not considered, as the prediction accuracy varies at the different locations along the ridge.

b) **For the towers inside the valley:** The valley is a zone of flow re-circulation and comprises lower velocities and higher variability, which remains challenging for prediction models. Moreover, large uncertainties are seen in the wind velocity, wind direction and turbulent kinetic energy profiles for the field measurements. The prediction capabilities of the models vary with the location of the tower inside the valley. At all towers inside the valley, the SF1 model (k-$\epsilon$) provides the best prediction for wind velocity. Most models show large relative errors in wind speed and turbulent kinetic energy profiles, especially close to the ground.

c) **For the towers on the Northeast ridge**: The region at the Northeast ridge is a zone of flow acceleration downstream of the re-circulation zone from the valley. The canopy model (SF3) provides a strong under-prediction, while all other models provide a prediction within one standard deviation of the field measurements. Predicting the extent of the re-circulation inside the valley and the re-attachment location plays a key role in the prediction profiles on the Northeast ridge. Significant turbulence is seen close to the ground in the field measurements and is under-predicted by most models.

d) **Influence of wind direction:** A significant difference in the wind profiles is seen using different inflow directions. The extent of the re-circulation zone and the re-attachment downstream of the valley is different due to different trajectories taken by the inflow wind profiles coming from the South-West ridge. These uncertainties also depend on the turbulence model and

315 source terms utilized.

The choice of the best-performing turbulence model is inconclusive in terms of overall prediction capability for different parts of the terrain. In the future, the surface heterogeneity of the canopy could be modeled based on the surface roughness length map at Perdigão. It would be more correct to use different patches with different heights, as seen in the forest point 320 cloud data. Simulations using non-uniform leaf area density could also be performed. Finally, an additional grid spacing study has to be done with a finer resolution as a part of future work.

*Acknowledgements.* The authors acknowledge the European Commission for its financial support through the project H2020-MSCA-ITN-2019 zEPHYR (Grant Agreement No. 860101). Perdigão data provided by NCAR/EOL under the sponsorship of the National Science Foundation. https://data.eol.ucar.edu/ The authors would like to thank Paul van der Laan for his useful insights, review, and comments which 325 helped improve the quality of the manuscript.

*Author contributions:* KV conceived, coordinated and was responsible for both the work, the manuscript writing, and the review process. TOH carried out the numerical simulations, manuscript writing, and preparation of figures, and attended in the review process. SB helped with conception, manuscript writing and review. KEG helped with model setup and review.

*Competing interests:* The authors declare that they have no conflict of interest.

*Data availability:* A community has been setup in Zenodo (https://zenodo.org/communities/zephyr/). A repository of the numerical setup shall be added through Github (https://github.com/kartikv95/WESC-Perdigao).

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

## Appendix

### Grid spacing study

The influence of grid spacing has been studied using five different grids as shown in Table 7. The "Very Fine" grid mesh is the maximum number of cells that could be generated due to computational constraints. The structured grids are generated using the `terrainBlockMesher` tool (Schmidt et al., 2012), which interpolates the SRTM terrain data and creates the terrain patch, which is blended into a cylindrical domain. All simulations are performed using the simpleFoam (SF1) solver with the $k - \epsilon$ turbulence model setup. The simulation inflow profiles for all the meshes are calibrated to match the field experiment at 100 m of Tower 20 (tse04) on the South-West ridge.

Significant differences are seen between the "Very Coarse" and "Medium" grids especially close to the ground near the Tower 20 (Calibration tower) on the South West ridge seen in Fig 23, inside the valley as seen in Fig. 24, and at Tower 29 close to 100 m on the North East ridge seen in Fig 25. Larger differences are seen close to the ground as the topology of the terrain is further resolved near the surface upon grid refinement. Increasing the number of cells refines and slightly modifies the surface mesh close to the ground. Furthermore, a small change in the prediction of the extent of the re-circulation zone could have a significant change and uncertainty in the predictions inside the valley and on top of the Northeast ridge. The order of these differences is similar to the order of the differences between the different model setups shown in the paper. This suggests that grid spacing is an important parameter for flow prediction around a complex terrain, and a grid-independent solution could be challenging to achieve based on the complexity of the flow topology. Further studies have to be performed with a finer grid spacing.

**Table 7.** Grid refinement study parameters showing the number of cells per main direction.

| Case | Nx | Ny | Nz | NCells (million) |
|---|---|---|---|---|
| **Very Coarse** | 300 | 300 | 80 | 12 |
| **Coarse** | 460 | 460 | 120 | 30 |
| Medium | 500 | 500 | 150 | 62 |
| Fine* | 550 | 550 | 170 | 88 |
| Very Fine | 650 | 650 | 190 | 115 |

### Inflow profiles - precursor and log-law

As shown in Fig 26, two sets of inflow profiles are utilized. An idealized set of inlet profiles, including a logarithmic velocity profile, and a fully developed profile using a precursor driven by a Coriolis force and a pressure gradient force. The inflow velocity profiles are calibrated to reach the desired inflow conditions at the met-mast Tower 20 at a height 100 m, for a time period identified as neutral based on the Bulk Richardson number. The wind velocity magnitude profiles are close to logarithmic. For the wind direction, the idealized profile fixes a uniform wind direction, but the precursor has a source term to account for the Coriolis effect, so a wind veer is seen over the entire height. Finally, for the TKE, a profile is set in the idealized

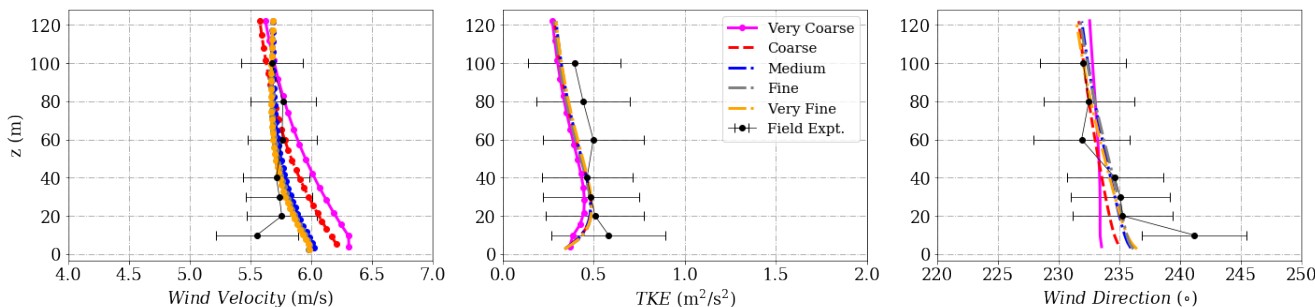

**Figure 23.** a) Wind velocity magnitude b) Wind direction c) Turbulent kinetic energy tuned to reach calibration at height 573 m corresponding to 100 m at Tower 20 (tse04)

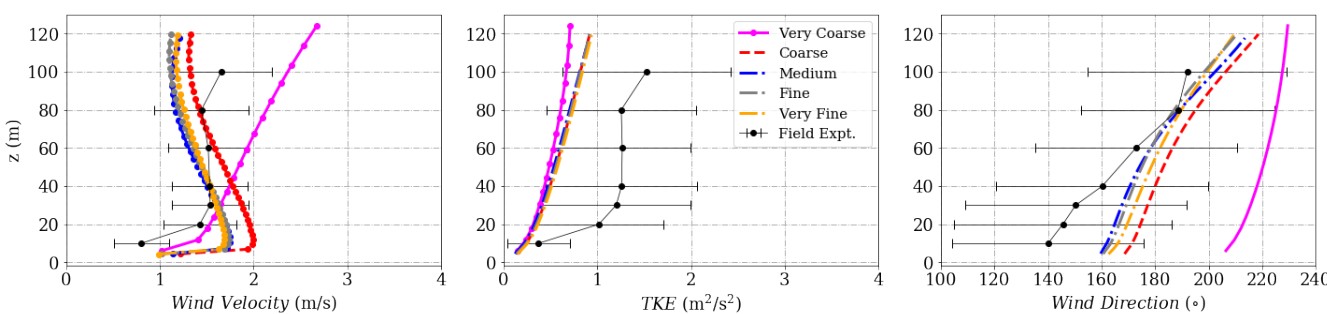

**Figure 24.** a) Wind velocity magnitude b) Wind direction c) Turbulent kinetic energy at Tower 25 (tse09)

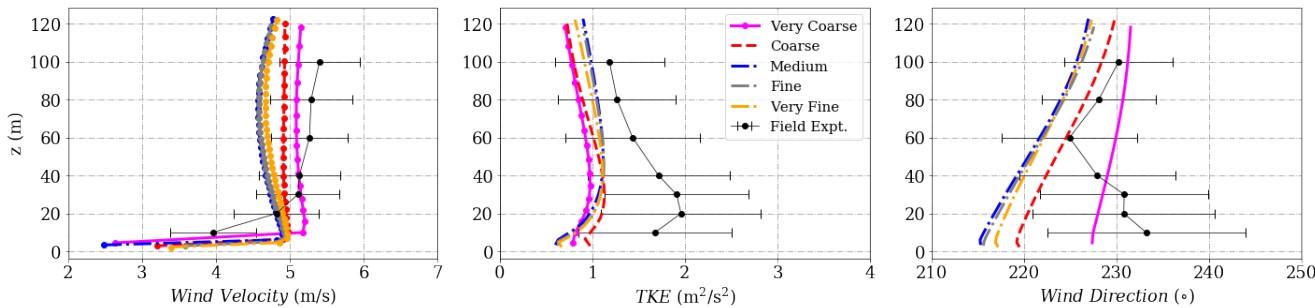

**Figure 25.** a) Wind velocity magnitude b) Wind direction c) Turbulent kinetic energy at Tower 29 (tse13)

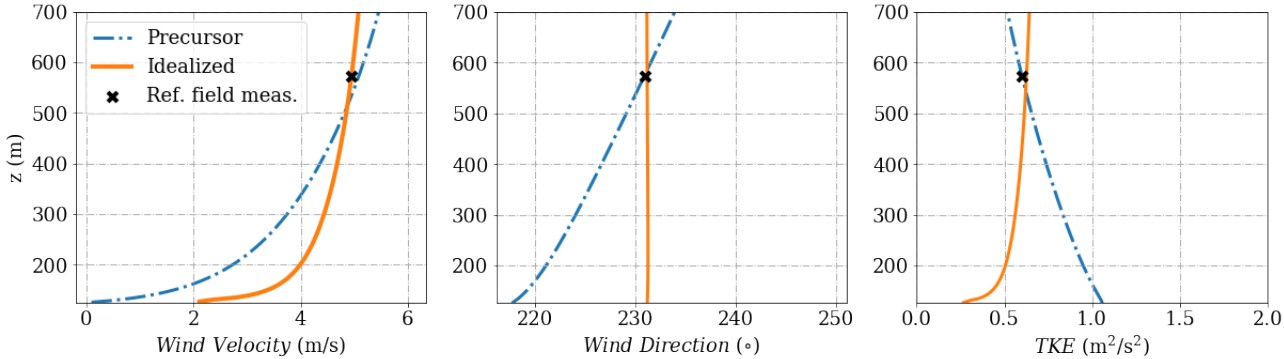

**Figure 26.** Idealized and Precursor (developed) input profiles for a) Wind velocity magnitude b) Wind direction c) Turbulent kinetic energy tuned to reach calibration at height 573 m corresponding to 100 m at Tower 20 (tse04)

case while for the precursor developed profile it is limited by the maximum mixing length scale. Increasing the roughness length z0 of the precursor to 6 (m), while also tuning pressure gradient magnitude and direction to get the right wind speed, wind direction, and now TKE at the reference height to match the meet the average field experiment value at the calibration point.

### Governing equations

The equations solved for the buoyantBoussinesqSimpleFoam solver are the continuity equation (Eqn. 3), momentum equation (Eqn. 4) and turbulence transport (Eqn. 6, 7), temperature (Eqn. 5) as described by Alletto et al. (2018). The pressure gradient ($\pi_i$) drives the momentum equation and the source terms for Coriolis and canopy effects are included in the momentum equation. The buoyancy terms are only added in the turbulence transport equations, but are set to zero for the present neutral case. The equation for turbulent dissipation rate contains the maximum length turbulence scale limiter ($l_{max}$) to modify the mixing-length scale estimations for setting different atmospheric stabilities. This description has been added to the Appendix of the manuscript.

$$\frac{\partial \rho \bar{u}_i}{\partial x_i} = 0 \tag{3}$$

$$\frac{\partial \rho \bar{u}_i}{\partial t} + \frac{\partial \rho \bar{u}_i \bar{u}_j}{\partial x_j} - \frac{\partial}{\partial x_j}\left(\rho(v + v_t)\left(\frac{\partial \bar{u}_i}{\partial x_j} + \frac{\partial \bar{u}_j}{\partial x_i}\right)\right) = -\frac{\partial \bar{p}}{\partial x_i} - \pi_i - \rho\epsilon_{ijk}f_j\bar{u}_k - \rho c_d \Sigma \bar{u}_i |\bar{u}| \tag{4}$$

$$\frac{\partial \rho \bar{\theta}}{\partial t} + \frac{\partial \rho \bar{u}_i \bar{\theta}}{\partial x_i} - \frac{\partial}{\partial x_i}\left(\rho\left(v + \frac{v_t}{\sigma_\theta}\right)\frac{\partial \bar{\theta}}{\partial x_i}\right) = S_\theta \tag{5}$$

$$\frac{\partial k}{\partial t} + \bar{u}_j \frac{\partial k}{\partial x_j} - \frac{\partial}{\partial x_j}\left(\left(v + \frac{v_t}{\sigma_k}\right)\frac{\partial k}{\partial x_j}\right) = P_k - \varepsilon + B \tag{6}$$

$$\frac{\partial \varepsilon}{\partial t} + \bar{u}_j \frac{\partial \varepsilon}{\partial x_j} - \frac{\partial}{\partial x_j}\left(\left(v + \frac{v_t}{\sigma_\varepsilon}\right)\frac{\partial \varepsilon}{\partial x_j}\right) = \left(C_{\varepsilon 1} + (C_{\varepsilon 2} - C_{\varepsilon 1})\left(\frac{l}{l_{\max}}\right)^a\right)\frac{\varepsilon}{k}P - C_{\varepsilon 2}\frac{\varepsilon^2}{k} +$$
$$((C_{\varepsilon 1} - C_{\varepsilon 2})\alpha_B)\frac{\varepsilon}{k}B - (C_{\varepsilon 1} - C_{\varepsilon 2})\frac{\varepsilon}{k}S_d \tag{7}$$

### Influence of Wind Direction using other models

### SF2 $k - \omega$

The influence of wind direction change using the SF2 ($k - \omega$) model at different towers is shown in Figs 27, 28 and 29 respectively. The results appear to be similar to the SF1 ($k - \epsilon$) model. However in comparison a lesser difference is seen in
the profiles between the $231°$ and the $234.5°$ degree cases at Towers 25 and 29.

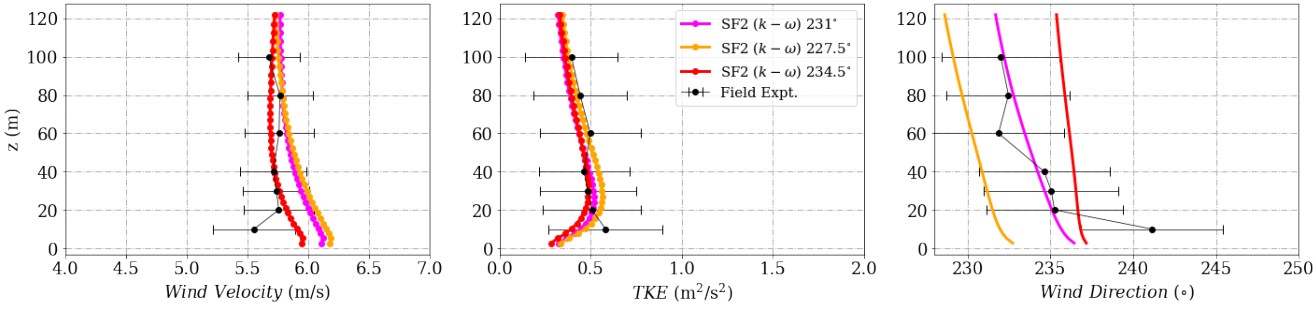

**Figure 27.** a) Wind velocity magnitude b) Wind direction c) Turbulent kinetic energy tuned to reach calibration at height 573 m corresponding to 100 m at Tower 20 (tse04) for the SF2 ($k - \omega$) model.

### SF3 Canopy

The influence of wind direction change using the SF3 (Canopy) model at different towers is shown in Figs 30, 31 and 32 respectively. A much larger difference in wind profiles is seen at Tower 29 on the North-East ridge compared to the SF1 ($k - \epsilon$) model indicating larger uncertainties with wind direction. Presently, the canopy is modeled using a uniform tree height, and
perhaps there would be higher uncertainties with a non-uniform canopy across the domain, the path taken by the wind for different inflows would vary significantly. A much different re-circulation zone is developed, as seen in the wind direction profiles.

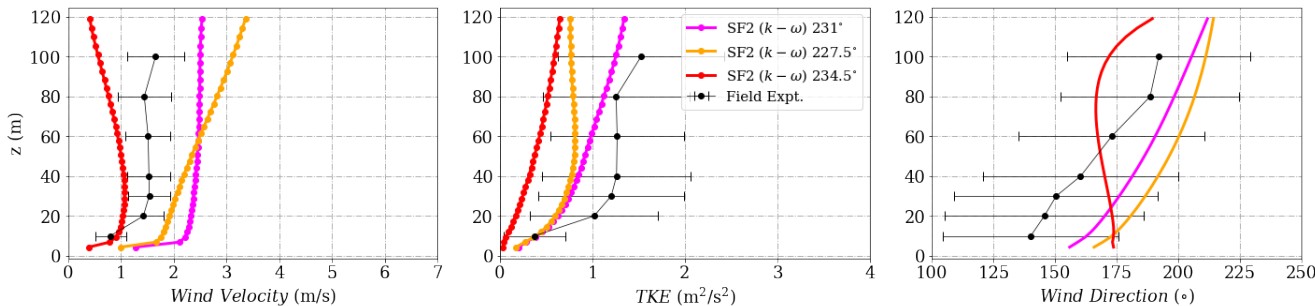

**Figure 28.** a) Wind velocity magnitude b) Wind direction c) Turbulent kinetic energy at Tower 25 (tse09) for the SF2 ($k - \omega$) model.

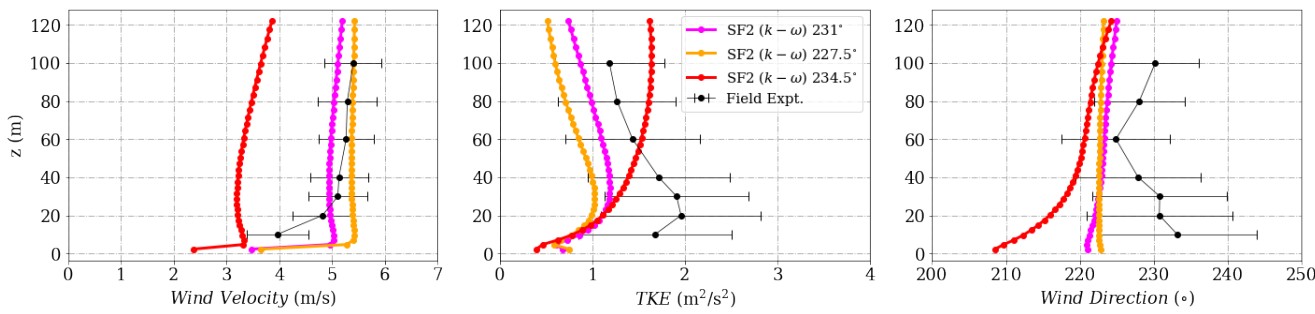

**Figure 29.** a) Wind velocity magnitude b) Wind direction c) Turbulent kinetic energy at Tower 29 (tse13) for the SF2 ($k - \omega$) model.

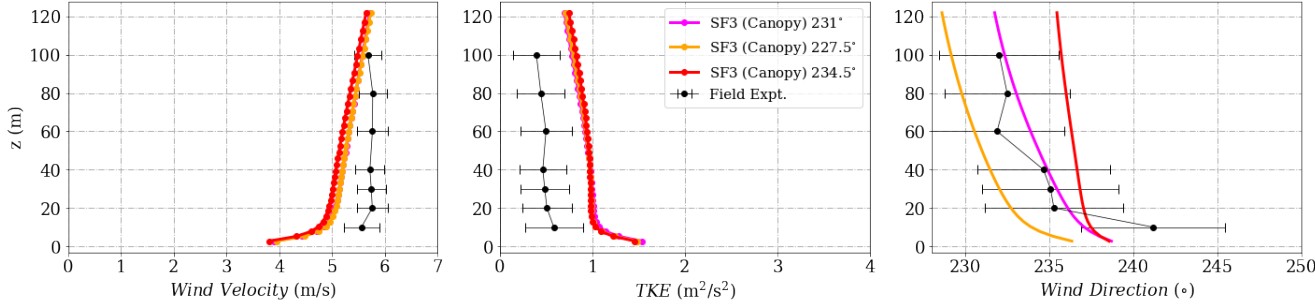

**Figure 30.** a) Wind velocity magnitude b) Wind direction c) Turbulent kinetic energy tuned to reach calibration at height 573 m corresponding to 100 m at Tower 20 (tse04) for the SF3 (Canopy) model.

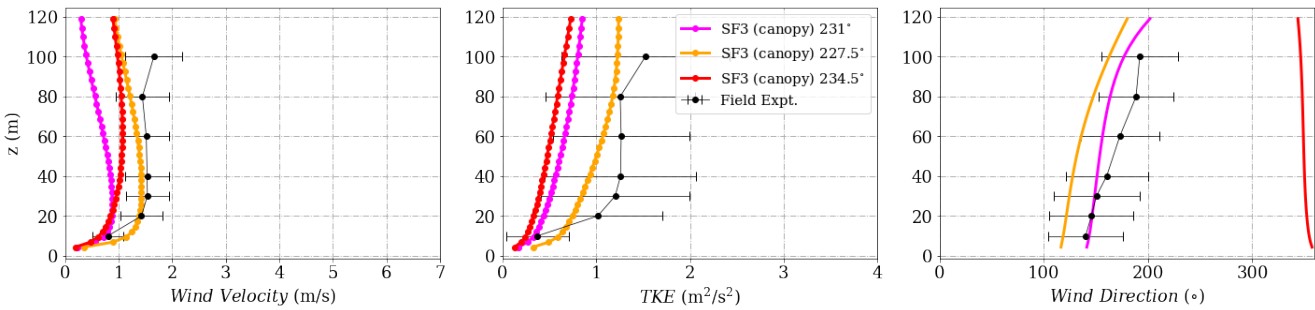

**Figure 31.** a) Wind velocity magnitude b) Wind direction c) Turbulent kinetic energy at Tower 25 (tse09) for the SF3 (Canopy) model.

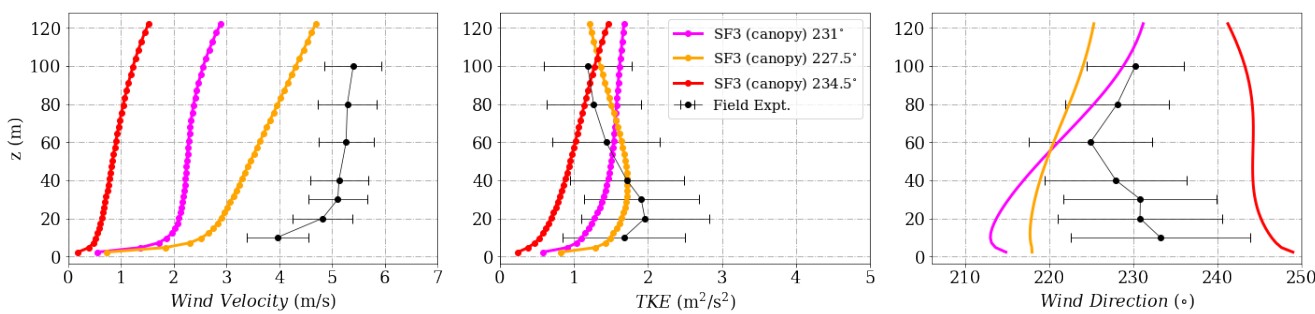

**Figure 32.** a) Wind velocity magnitude b) Wind direction c) Turbulent kinetic energy at Tower 29 (tse13) for the SF3 (Canopy) model.

## SF4 Coriolis

The influence of wind direction change using the SF3 (Coriolis) source term model at different towers is shown in Figs 33, 34 and 35 respectively. Interestingly, for this case, the flow difference between the $231°$ and the $234.5°$ degree cases at Towers 25 and 29 is relatively small compared to the SF1 $k - \epsilon$ case. With the Coriolis source term, the flow turning due to the Coriolis effect is accounted for, and hence the inflow wind for calibration is set accordingly to obtain the required wind direction at the calibration mast. Consequently, the path taken by the wind across the terrain is different.

## BBSF

Figs 36,37 and 38 show the wind profiles for the BBSF simulation model with the precursor inflow. Here the inflow from wind direction $234.5°$ shows a larger difference between the bother profiles even at the calibration mast, and a decrease in wind velocity at Tower 29 on the North-East Ridge, similar to the other models. The wind direction standard deviation bounds in Fig 38 appear to be more closely predicted by this model.

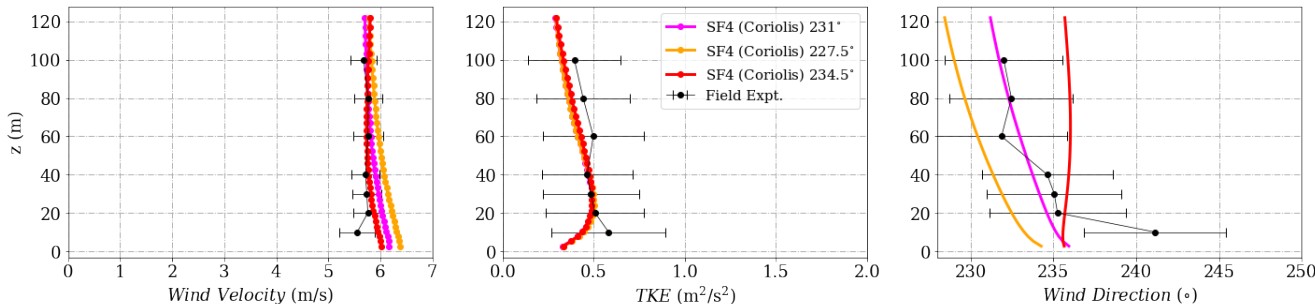

**Figure 33.** a) Wind velocity magnitude b) Wind direction c) Turbulent kinetic energy tuned to reach calibration at height 573 m corresponding to 100 m at Tower 20 (tse04) for the SF4 (Coriolis) model.

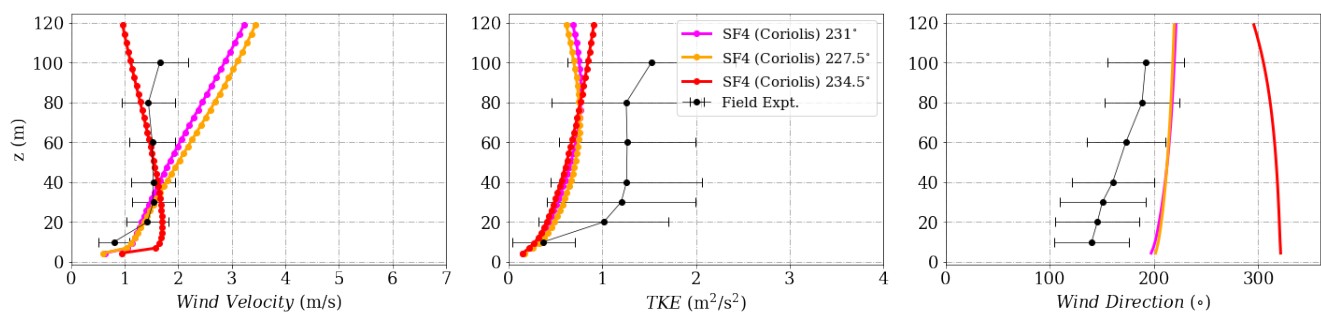

**Figure 34.** a) Wind velocity magnitude b) Wind direction c) Turbulent kinetic energy at Tower 25 (tse09) for the SF4 (Coriolis) model.

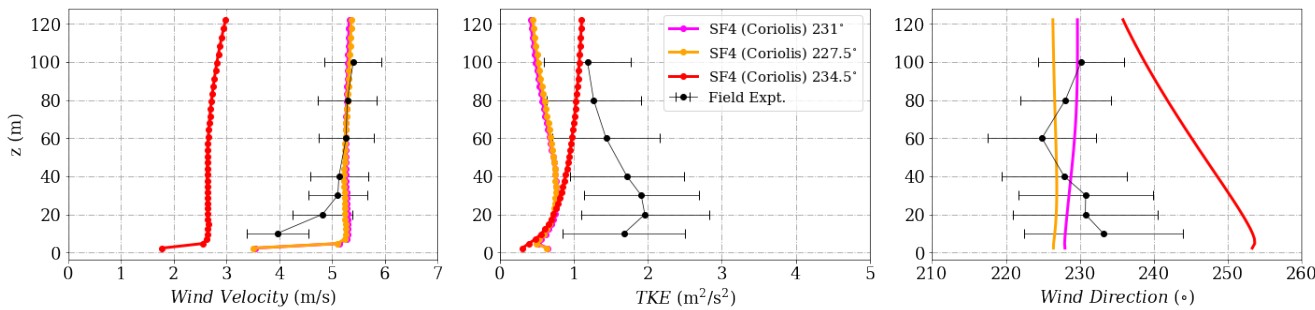

**Figure 35.** a) Wind velocity magnitude b) Wind direction c) Turbulent kinetic energy at Tower 29 (tse13) for the SF4 (Coriolis) model.

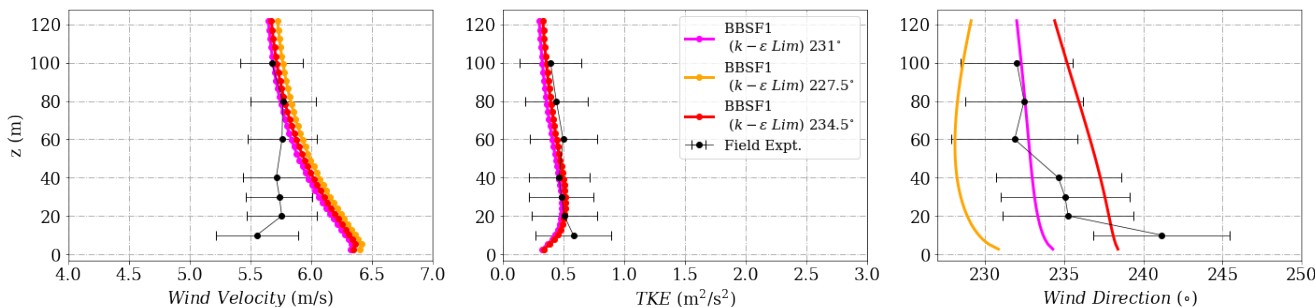

**Figure 36.** a) Wind velocity magnitude b) Wind direction c) Turbulent kinetic energy tuned to reach calibration at height 573 m corresponding to 100 m at Tower 20 (tse04) for the BBSF model.

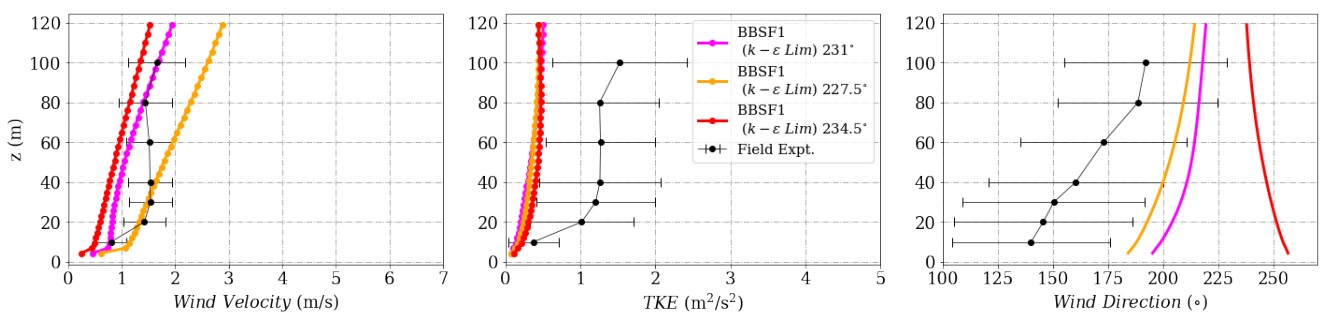

**Figure 37.** a) Wind velocity magnitude b) Wind direction c) Turbulent kinetic energy at Tower 25 (tse09) for the BBSF model.

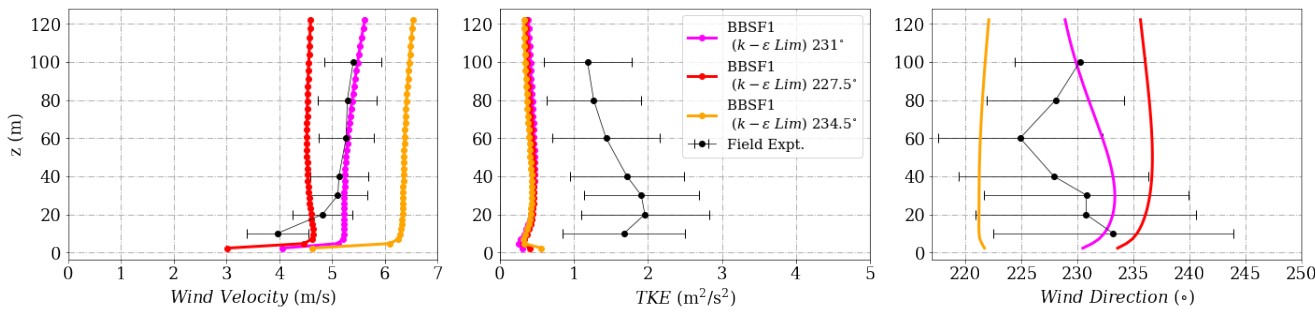

**Figure 38.** a) Wind velocity magnitude b) Wind direction c) Turbulent kinetic energy at Tower 29 (tse13) for the BBSF model.