# Peer review of "Effect of different source terms and inflow direction in atmospheric boundary modeling over the complex terrain site of Perdigao"

_Wind Energy Science, 2021_

## Referee Comment (RC1)

**Review of *Effect of different source terms in atmospheric boundary modelling over the complex terrain site of Perdigao* by K. Venkatraman et al.**

Reviewer: M. Paul van der Laan, DTU Wind Energy

February 17, 2022

The authors compare and validate results of different atmospheric RANS turbulence models applied to a complex terrain site.

I like the idea to investigate the effect of different atmospheric physics and the effect of a forest representation on complex terrain using two-equation RANS turbulence models. The topic is also well suited for Wind Energy Science. However, the article is not entirely clear because the RANS turbulence/inflow models and input parameters are not fully described. I think it is essential that you do describe them completely in order to be able to understand the article and provide the reader an opportunity to redo the simulations. More detailed comments are listed below; they need to be addressed before the article can be considered for publication in Wind Energy Science.

**Main comments**

1. Koblitz et al. (2013) used a range of turbulence models that differ in complexity and it is not entirely clear which elements you have adopted. For example, do you use an active temperature equation or do you only use the global turbulence length scale limiter of Apsley and Castro (1997) [1]? Do you use ambient source terms to avoid zero turbulence values above the ABL (see for example van der Laan (2020))? I propose that you write down the full model description of the momentum equations including possible source terms, the Boussinesq hypothesis and the $k$-$\varepsilon$ (and $\omega$) turbulence transport equations (also including all possible source terms). Then you can write in Table 2 which source terms are active by referring to the variable name ($S_?$). You could have a look at a recent article of my own where I tried to do this [3]. In addition, I strongly recommend to add a table including the chosen values of all turbulence model constants. Furthermore, not all parameters are defined. For example, what are the chosen values of $G$, $f_c$, $\beta_B$, $\alpha_t0$, etc? What was the set inflow wind direction (at a certain reference height)?

2. You use sources of buoyancy in the BBSF1 model while you are only considering a neutral case. Are these sources then set to zero? If this is the case, wouldn't it make sense to remove them from the article and also remove the word buoyancy in the abstract and elsewhere, since you have not yet investigate its effect?

3. Are you aware that the global turbulence length scale limiter of Apsley and Castro (1997) [1] can have problems when it is applied to complex terrain where the turbulence length scales of the hills are in the order of the maximum value that is set by turbulence model? When this is case one could observe non-physical large hill wakes or even numerical convergence problems. I think this is worth mentioning in the article. (Unfortunately, I couldn't find a good reference for it other then a brief discussion in an article of my own [4]).

4. Abstract, Line 10: I would rewrite the following sentence: *The inclusion of a canopy model is shown to improve predictions close to the ground for most of the towers, while reducing prediction accuracy on top of the ridges, illustrating the need to represent terrain heterogeneity.*, because the second point is not in favor of representing terrain heterogeneity.

5. Section 3: You mention that a grid refinement study was performed, but I could not find any results in the article. Also note that a reference to the grid refinement study of Laginha Palma (2020) is not sufficient because they have used a different solver and setup. (A grid refinement study is solver dependent.) In addition, the chosen turbulence model might also influence the grid study; you could show results of a grid refinement study using the most demanding turbulence model.

6. You forgot to add results of the inflow (precursor). It would be useful to compare the two inflow models in a plot for wind wind speed, wind direction, TKE, turbulence length scale and temperature.

7. Do you have an idea how much the wind direction varies in the observations? If this is significant you might need to account for it in the models by running a set of wind directions and then you can average the results (profiles) using a weighted averaged following a Gaussian distribution with a standard deviation representing wind direction uncertainty. A typical value for wind farm wake studies is $5°$ but it depends on the site and the distance between the location at which the reference wind direction was measured and the location at which the profile was measured. For more info you can have a look at Gaumond et al. (2013) [2] and a work of my own [5] (Section 3.1.3). For complex terrain, the effect of wind direction is often quite significant and a small difference in wind direction (distribution) between the measurements and models can result in large differences.

8. Line 115: You mention that you use a loglaw inflow with TKE profile that varies with height following a modification of Yang et al. (2009). If you want a varying TKE profile with height then you could just model a pressure-driven boundary layer with a constant pressure gradient, which should result in logarithmic wind speed profile near ground and a varying TKE profile. Such a model would require a precursor simulation to generate the inflow. In addition, I am not convinced that the model of Yang et al. (2009) is a solution of the standard $k$-$\varepsilon$ model, meaning that the inflow will most likely develop downstream (especially of the domain is large). Have you checked this?

9. Line 134: You write *The direction of the flux automatically determines the inlet and outlet regions.* I thought that the set wind direction would determine the inlet and outlet regions, or this is a misunderstanding from my side? How do you handle wind veer for determining the inlet and outlet boundaries?

10. Section 3.2, Line 144: What is the reason that you use a constant leaf area density? You could easily use a varying leaf area density based on the forest point cloud data.

11. Section 3.2, eq. (3): Is this really how the Coriolis force source term is implemented in your model? If you follow the ABL model of Koblitz et al. (2013) I would expect that you include $U$ and $V$-momentum source terms that represent a balance between a (constant) geostrophic wind speed and the Coriolis force (see for example van der Laan (2020)), as you also briefly discuss.

12. Section 3.2, eqns (2) and (4): I think you need to define different source terms, for example $S_{p,m}$, $S_{p,k}$ and $S_{p,\varepsilon}$.

13. Table 2: The skipped entries (–""—) are unclear to me. For example, does the BBSF1 case include forest source terms or not? I would just fill in the entire table for clarity. In addition, what do you mean by using SKE for BBSF1? I though that you use a global length scale limiter in the epsilon equation, which is different from a standard model.

14. What do the error bars on the measurements represent? Is is the standard deviation of the uncertainty of the mean? I would recommend to use the latter.

15. You could group the profile plots in the results into three figures, where each row of sub plots represents a mast: SW ridge (combine Figs. 6-8), valley (combine Figs. 9-12), NE ridge (combine Figs. 13-14). I think this make sense because you also discuss them as met mast groups in the text. In addition, you could consider to plot normalized results of wind

speed ($U/U_\text{ref}$), turbulence intensity ($\sqrt{2/3k}/U_\text{ref}$) and wind direction ($wd - wd_\text{ref}$) instead of dimensional results. Finally, you could zoom the $x$-axis of the wind direction, since it is hard to see the difference between the models and measurements in Figs. 6c, 7c, 8c ad 14c. This also applies to some of the wind speed and TKE plots.

16. Line 194: You mention: *A good match is obtained in between the measured and computed velocity, turbulent kinetic energy and wind direction profiles for the calibration Tower 20 as seen in Fig. 6.* However, the results of the model including forest are not matching well, especially for the TKE and the wind speed near the ground.

17. Figure 13: Nice plot. You could add the relevant mast location(s) in the plot, so the reader can better understand the results of Figs. 9-12.

18. I am missing information on code and data availability, which is normally added at the end of the article. In addition, I was wondering if is possible to provide the numerical setup using a DOI of a git hub repository through Zenodo or something similar. By proving the numerical setup/ run scripts, one could easily redo the work since the numerical solver (OpenFOAM) is publicly available.

**Minor comments**

1. Introduction, Line 13: You could rewrite the following *Lack of terrain availability in flat terrain pushes wind-farm developers to look for alternative sites along complex terrains.*, since you use the word terrain three times and I think complex terrains could be rewritten as complex terrain sites. The latter also applies elsewhere in the paper.

2. Line 117: log-low $\rightarrow$ log-law.

3. There are quite a lot of other typos in the article but these can be fixed in the proof reading process.

**References**

[1] Apsley, D. D. and Castro, I. P. A limited-length-scale $k$-$\varepsilon$ model for the neutral and stably-stratified atmospheric boundary layer. *Boundary-Layer Meteorology*, 83:75, 1997.

[2] Gaumond, M., Réthoré, P.-E., Ott, S., Peña, A., Bechmann, A., and Hansen, K. S. Evaluation of the wind direction uncertainty and its impact on wake modeling at the Horns Rev offshore wind farm. *Wind Energy*, 17(8):1169, 2014.

[3] van der Laan, M. P., Baungaard, M., and Kelly, M. Inflow modeling for wind farm flows in RANS. *Journal of Physics: Conference Series*, (1):1–11, 2021.

[4] van der Laan, M. P., Kelly, M., and Baungaard, M. A pressure-driven atmospheric boundary layer model satisfying rossby and reynolds number similarity. *Wind Energy Science*, 6(3):777–790, 2021.

[5] van der Laan, M. P., Sørensen, N. N., Réthoré, P.-E., Mann, J., Kelly, M. C., Troldborg, N., Hansen, K. S., and Murcia, J. P. The $k$-$\varepsilon$-$f_p$ model applied to wind farms. *Wind Energy*, 18(12):2065, December 2015.

---

## Referee Comment (RC2)

**Review of "Effect of different source terms in atmospheric boundary modelling over the complex terrain site of Perdigão" by K. Venkatraman et al.**

*José Carlos P. Lopes da Costa, ISEP*

*April 7, 2022*

This article present the results of different simulations of the wind behaviour on Perdigão site, comparing it with the results of a field measuring campaign on this region of complex terrain. The authors used several combinations of two-equation RANS turbulence models with a forest canopy model and atmospheric stratification conditions. I think that this is a quite interesting subject and very adequate to be published on a wind energy publication.

However, in order to be considered for publication, I think that this article lacks of a more detailed description of the implementation of the different models used; furthermore, it will be necessary to better specify the parameters used in the different numerical models, such as the forest canopy model, so that they can be eventually mimicked in other studies.

**Comments**

1. From reading the article, one can not know precisely what TKE models parameters were used in the simulations. As an example, the $k - \varepsilon$ model can use different model constant values such as $C_\mu$ used in the calculation of the turbulent viscosity $\mu_T = \rho C_\mu \dfrac{k^2}{\varepsilon}$. Is is usually used the standard value $C_\mu = 0.090$; however, for atmospheric flows $C_\mu = 0.033$ seams to be more adequate. Is that the case? There are also other parameters that are not revealed, such as the Prandtl number for the turbulent dissipation and others. Do you use the default values in OpenFOAM? Even if it is the case, it should be enumerated.

2. A similar problem appear in the canopy model description; the model parameters aren't completely defined. It is said that the model is based in [Lopes da Costa, 2007], but this model uses several parameters that are not fully addressed in this article. It is presented the source term for the velocity and one can not know what is the $\alpha$ parameter in it or what that represents. The source/sink terms for the turbulent kinetic energy and its dissipation rate are not also mentioned. I think that you might be more clear in this subject.

3. In chapter 3, though the definition of the domain volume is well explained, the description of the computational mesh is vague. It

is said that it consists in 12.7 million cells, but it would be more useful to define it by the number of cells per main direction, such as $n_x \times n_y \times n_z$. It is said that the horizontal mesh resolution was set to 33 m, but is this resolution observed only in the center of the domain (close to the masts location) expanding itself to the boundaries or is it a regular mesh? (...which is probably not the case, as the domain is not square or rectangular, but cylindrical.) It is also said that an "uniform stretching is applied to the vertical direction" with no more details. However, one of the most important mesh parameters in an atmospheric flow simulation is the minimal mesh hight $\Delta z$ next to the ground (along with the vertical number of cells), which is not presented in the article.

4.  It is said (line 145) that you have choose a mean tree heigh of 3 m. I suppose that it is all over the domain (or, at least in the $7,5$ km $\times$ 7.5 km squared area - it is not clear in the article. Wouldn't it be more correct to use different patches (eventually with different heights), as you have that information in figure 5? Or, at least remove the trees from the higher zones of the ridges, as it is common to happen in this kind of topography and cam be observed in figure 5?

5.  I suppose that you use in the canopy model an uniform area density. However, forests have a higher foliage density at the half top than at the bottom zones - [Lalic and Mihailovic, 2004]. I think a non-uniform leaf area density could be easily implemented in that model.

6.  I find the Figure 13 very interesting. The extension of the slice could be wider, in order to include the whole top of the ridges and clearly show the positions of the masts 10 and 7 (and approximately the zone of mast 37). This small change could complement and be more enlightening in the interpretation of the results obtained for these masts and presented in the other figures. (Also could be useful for the reader if the caption of the figure refer that the location of slice is defined in Figure 2.)

**Minor Comments**

1.  Line 111: "...homogeneous atmospheric boundary layer (ABL). either..." should be "...homogeneous atmospheric boundary layer (ABL) either...".

2.  Figure 5: The scale in the figure is strange; the sequence of the $d$ (m) values is $0, 2, 6, 4, 8$; shouldn't it be $0, 2, 4, 6, 8$?

3. For example: In Figure 5, m for meters shouldn't be written in italic as it is not a variable, but a length unit. In many other figures there is also the variables (*U*, *TKE*,...) not in italic, but with the units in italic; it should be the opposite.

4. Table 2: I would prefer the description of each case to be done in a more clear way, instead of using " –"– " all over the table.

5. Line 111: "predictions" is two times written.

6. Line 205 to 206: I think that the order of the towers and the figures is somehow messed up. It should be "...towers 25, 7, 27 and 22.", and further, "...shown in Figs. 9, 10, 11 and 12 respectively.".

7. ...and a small detail: in the references, "Costa, J. L. C. (2007)" should be "Costa, J. C. L. (2007)", or even "Lopes da Costa, J. C. (2007)". ;)

**References**

Branislava Lalic and Dragutin Mihailovic. An empirical relation describing leaf-area density inside the forest for environmental modeling. *Journal of Applied Meteorology - J APPL METEOROL*, 43: 641–645, 04 2004.

J. C. Lopes da Costa. *Atmospheric flow over forested and non-forested complex terrain*. PhD thesis, University of Porto, 2007.

---

## Author Comment (AC1)

**Response to the reviewer**

We thank the reviewer #1 for their useful comments and the time invested in reviewing our manuscript. We have addressed each of the referee comments as detailed point by point below, which we believe has significantly improved the quality of the manuscript.
* * *
**Reviewer 1**

**Main Comments**

**Reviewer Point P 1.1** — Koblitz et al. (2013) used a range of turbulence models that differ in complexity and it is not entirely clear which elements you have adopted. For example, do you use an active temperature equation or do you only use the global turbulence length scale limiter of Apsley and Castro (1997) [2]? Do you use ambient source terms to avoid zero turbulence values above the ABL (see for example van der Laan (2020))? For example, what are the chosen values of $G, f_c, \beta_B, \alpha_{t0}$, etc? What was the set inflow wind direction (at a certain reference height)? propose that you write down the full model description of the momentum equations including possible source terms, the Boussinesq hypothesis and the $k - \varepsilon$ (and $\omega$) turbulence transport equations (also including all possible source terms). Then you can write in Table 2 which source terms are active by referring to the variable name ($S_?$). You could have a look at a recent article of my own where I tried to do this [6]. In addition, I strongly recommend to add a table including the chosen values of all turbulence model constants. Furthermore, not all parameters are defined.

**Reply**: We agree with the reviewer on this important point on clear explanation of the different elements adopted in the turbulence models. Table 3 has been added in the Methodology section of the manuscript detailing all the different turbulence model constants.

   More specific details with regards to the buoyantBoussinesqsimpleFoam solver (BBSF) model, which uses a precursor developed inflow, is described as follows. For the BBSF model, Ambient source terms was added to avoid zero turbulence values above the ABL as mentioned in [6] and applied to the entire domain. The values were set to $k_{Amb} = 0.001$ and $\epsilon_{Amb} = 7.208e - 08$ as summarized in Table 3 in the manuscript. The global turbulence length scale limiter of Apsley and Castro (1997) [2] is utilized and an active temperature equation is also solved.

   The Coriolis force source term is included as a momentum source term ($U$ and $V$ momentum equations), this has been corrected in the manuscript in the Methodology Section 3.2. The Coriolis force is calculated based on the planetary rotational period ($\Omega = 24\,h$) and the latitude ($\lambda$) for Perdigao (39.68 $°N$).

   The inflow wind direction at the inlet at a reference height was set to calibrate the model at a reference Tower 20, at a height of 100 m above the ground which is 573 m above sea level. The inflow profiles are shown in response to Question 6 P 1.6.

   The equations solved for the buoyantBoussinesqSimpleFoam solver are the continuity equation (Eqn. 1), momentum equation (Eqn. 2) and turbulence transport (Eqn. 4, 5), temperature (Eqn. 3) as described by Alletto *et al.* [1]. The pressure gradient ($\pi_i$) drives the momentum equation, the source terms for Coriolis and Canopy effects are included in momentum equation. The buoyancy terms are only added in the turbulence transport equations, but are set to zero for the present neutral case. The equation for turbulent dissipation rate contains the maximum length turbulence scale limiter ($l_{max}$) to modify the

mixing-length scale estimations for setting different atmospheric stabilities. This description has been added to the Appendix of the manuscript.

$$\frac{\partial \rho \bar{u}_i}{\partial x_i} = 0 \tag{1}$$

$$\frac{\partial \rho \bar{u}_i}{\partial t} + \frac{\partial \rho \bar{u}_i \bar{u}_j}{\partial x_j} - \frac{\partial}{\partial x_j}\left(\rho\left(v + v_t\right)\left(\frac{\partial \bar{u}_i}{\partial x_j} + \frac{\partial \bar{u}_j}{\partial x_i}\right)\right) = -\frac{\partial \bar{p}}{\partial x_i} - \pi_i - \rho \epsilon_{ijk} f_j \bar{u}_k - \rho c_d \Sigma \bar{u}_i |\bar{u}| \tag{2}$$

$$\frac{\partial \rho \bar{\theta}}{\partial t} + \frac{\partial \rho \bar{u}_i \bar{\theta}}{\partial x_i} - \frac{\partial}{\partial x_i}\left(\rho\left(v + \frac{v_t}{\sigma_\theta}\right)\frac{\partial \bar{\theta}}{\partial x_i}\right) = S_\theta \tag{3}$$

$$\frac{\partial k}{\partial t} + \bar{u}_j \frac{\partial k}{\partial x_j} - \frac{\partial}{\partial x_j}\left(\left(v + \frac{v_t}{\sigma_k}\right)\frac{\partial k}{\partial x_j}\right) = P_k - \varepsilon + B \tag{4}$$

$$\frac{\partial \varepsilon}{\partial t} + \bar{u}_j \frac{\partial \varepsilon}{\partial x_j} - \frac{\partial}{\partial x_j}\left(\left(v + \frac{v_t}{\sigma_\varepsilon}\right)\frac{\partial \varepsilon}{\partial x_j}\right) = \left(C_{\varepsilon 1} + (C_{\varepsilon 2} - C_{\varepsilon 1})\left(\frac{l}{l_{\max}}\right)^a\right)\frac{\varepsilon}{k}P - C_{\varepsilon 2}\frac{\varepsilon^2}{k} +$$
$$((C_{\varepsilon 1} - C_{\varepsilon 2})\alpha_B)\frac{\varepsilon}{k}B - (C_{\varepsilon 1} - C_{\varepsilon 2})\frac{\varepsilon}{k}S_d \tag{5}$$

**Reviewer Point P 1.2** — You use sources of buoyancy in the BBSF1 model while you are only considering a neutral case. Are these sources then set to zero? If this is the case, wouldn't it make sense to remove them from the article and also remove the word buoyancy in the abstract and elsewhere, since you have not yet investigate its effect?

**Reply**: We agree with the reviewer on this point. Indeed the surface heating on the ground was set to zero, as the simulation period can be assumed as a near-neutral condition. We have instead renamed the models as a idealized (log-law) and a precursor inflow model. The word buoyancy has been removed from the article in the abstract and elsewhere, since we have not yet investigated its effect.

**Reviewer Point P 1.3** — Are you aware that the global turbulence length scale limiter of Apsley and Castro (1997) [1] can have problems when it is applied to complex terrain where the turbulence length scales of the hills are in the order of the maximum value that is set by turbulence model? When this is case one could observe non-physical large hill wakes or even numerical convergence problems. I think this is worth mentioning in the article. (Unfortunately, I couldn't find a good reference for it other then a brief discussion in an article of my own [7].)

**Reply**: Indeed, we encountered numerical convergence difficulties when using the global turbulence length scale limiter when applied to the present case for simulations over the complex terrain of Perdigão. This is especially true when setting the values of the maximum limiting length scale to low values. Here, the model is trying to restrict the turbulence scale, but physically the large length scales of turbulence are produced from hills or features that are in the order of the maximum value that is set by turbulence model. This has been mentioned in the article in Section 3 at lines 195-200.

**Reviewer Point P 1.4** — Abstract, Line 10: I would rewrite the following sentence: The inclusion of a canopy model is shown to improve predictions close to the ground for most of the towers, while reducing prediction accuracy on top of the ridges, illustrating the need to represent terrain heterogeneity., because the second point is not in favor of representing terrain heterogeneity.

**Reply**: This sentence has been re-written in the Abstract, we thank the reviewer for the suggestion. The inclusion of a canopy model is shown to improve predictions close to the ground for the towers on the South-West ridge and inside the valley. However the model, performs poorly for predictions on the North-East ridge.

**Reviewer Point P 1.5** — Section 3: You mention that a grid refinement study was performed, but I could not find any results in the article. Also note that a reference to the grid refinement study of Laginha Palma (2020) is not sufficient because they have used a different solver and setup. (A grid refinement study is solver dependent.) In addition, the chosen turbulence model might also influence the grid study; you could show results of a grid refinement study using the most demanding turbulence model.

**Reply**:

A mesh refinement study has been performed as shown in Table 1, by increasing the number of cells in the x and y direction thereby increasing the horizontal mesh resolution. The case has been simulated using the standard $k - \epsilon$ turbulence model with an idealized log-law inflow and no Coriolis source term.

Table 1: Grid refinement study

| Case | Nx | Ny | Nz | NCells (million) |
|---|---|---|---|---|
| Coarse | 227 | 227 | 120 | 12.7 |
| Medium | 332 | 332 | 120 | 21.19 |
| Fine | 469 | 469 | 120 | 30.6 |

Negligible differences are seen in the velocity and turbulent kinetic energy profiles on top of the ridge at the Tower 20 (Calibration tower) on the South West ridge seen in Fig 1 and at Tower 29 close to 100 m on the North East ridge seen in Fig 2. However, larger differences are seen close to the ground as the topology of terrain is further resolved near the surface upon grid refinement. The structured grids are generated using the `terrainBlockMesher` tool [11], which interpolates the SRTM terrain

data and creates the terrain patch which is blended into a cylindrical domain. Increasing the number of cells refines and slightly modifies the surface mesh close to the ground. Similar differences are seen, when using different terrain databases in the grid refinement study by Palma *et al.* [9]. Furthermore, a small change in the prediction of extent of the re-circulation zone could have a significant change and uncertainty in the predictions inside the valley in Fig 3 on top of the North-East ridge.

[Figure]

Figure 1: Profiles for a) Velocity Magnitude b) Turbulent kinetic energy c) Meteorological wind direction at Tower 20 (tse04) on the South West ridge, shown for different levels of grid refinement.

[Figure]

Figure 2: Profiles for a) Velocity Magnitude b) Turbulent kinetic energy c) Meteorological wind direction at Tower 29 (tse13) on the North East ridge, shown for different levels of grid refinement

[Figure]

Figure 3: Profiles for a) Velocity Magnitude b) Turbulent kinetic energy c) Meteorological wind direction at Tower 25 (tse09) inside the valley, shown for different levels of grid refinement

**Reviewer Point P 1.6** — You forgot to add results of the inflow (precursor). It would be useful to compare the two inflow models in a plot for wind wind speed, wind direction, TKE, turbulence length scale and temperature.

**Reply**:  Thank you for your suggestion, and we agree that this information is useful for the readers. Therefore we have added figures of the inflow profiles for both the sets of inlet profiles used in the study in the Appendix.

Two sets of inflow profiles are utilized as shown in Fig 4- an idealized log-law profile for wind velocity, and a developed profile using a precursor driven by a pressure gradient. The inflow velocity profiles are calibrated to reach the desired inflow conditions at the met-mast Tower 20 at a height 100 m, for a time period identified as neutral based on the bulk Richardson number. The wind velocity magnitude profiles is close to logarithmic. For the wind direction, the idealized profile fixes a uniform wind direction, but the precursor has source term to account for the Coriolis effect, so a wind veer is seen over the entire height. Finally, for the TKE, a profile is set in the idealized case while for the precursor developed profile it is limited by the maximum mixing length scale, hence there is a decrease in turbulence levels over height.

The temperature plots are not shown as this is a neutral case with the source term for buoyancy not activated.

[Figure]

Figure 4: Idealized and Precursor (developed) input profiles for a) Wind velocity magnitude b) Wind direction c) Turbulent kinetic energy tuned to reach calibration at height 573 m corresponding to 100 m at Tower 20 (tse04)

**Reviewer Point P 1.7** — Do you have an idea how much the wind direction varies in the observations? If this is significant you might need to account for it in the models by running a set of wind directions and then you can average the results (profiles) using a weighted averaged following a Gaussian distribution with a standard deviation representing wind direction uncertainty. A typical value for wind farm wake studies is 5° but it depends on the site and the distance between the location at which the reference wind direction was measured and the location at which the profile was measured. For more info you can have a look at Gaumond et al. (2013) [4] and a work of my own [8] (Section 3.1.3). For complex terrain, the effect of wind direction is often quite significant and a small difference in wind direction (distribution) between the measurements and models can result in large differences.

**Reply**:  Yes, the instantaneous wind direction within the 30-min interval has a standard deviation of around 7 degrees as seen in Fig 5 (Figure 8 in the manuscript). This uncertainty is also dependent on location and height, for example there is a very high variance for Tower 34 close to the ground as expected for a complex terrain. We agree that simulating a set of wind directions measured within the time-period and then performing some statistics on the data would likely provide a more accurate representation of the conditions in the period. However, we have decided not to do this as this is not

the scope of our article. Our scope is to study the effect of the different source terms on the wind simulations for only one inlet wind direction (with or without veer), which represents the conditions in a 30-min interval. This highlight limitations in the models' ability to represent actual conditions for such a complex site. Therefore, comments in the discussion part have been added in the Results.

There have been studies performed on the effect of inflow wind direction [4, 8]. . As a part of future work, an uncertainty quantification using the tool DAKOTA similar to the strategy adopted by Garcia-Sanchez *et al.* [3] can be performed, with an input matrix of wind directions and wind velocity uncertainty intervals. The range of inflow variability accounted for, needs to be less than the variability in the field measurements, an inherent variability in the inflow wind direction is already included in the turbulence model as shown by Vervecken *et al.* [12].

[Figure]

Figure 5: Simulation results and experimental data for wind direction on the South-West ridge for a) tower 20/ tse04 b) Tower rsw03 c) Tower rsw06

**Reviewer Point P 1.8** — Line 115: You mention that you use a log-law inflow with TKE profile that varies with height following a modification of Yang et al. (2009). If you want a varying TKE profile with height then you could just model a pressure-driven boundary layer with a constant pressure gradient, which should result in logarithmic wind speed profile near ground and a varying TKE profile. Such a model would require a precursor simulation to generate the inflow. In addition, I am not convinced that the model of Yang et al. (2009) is a solution of the standard $k$ - $\varepsilon$ model, meaning that the inflow will most likely develop downstream (especially of the domain is large). Have you checked this?

**Reply**:

The use of both the Idealized (log-law) inflow and a precursor inflow model has been done. The inflow profile does develop downstream, however since the domain is cylindrical with a smoothing region applied and since it it is not flat, we indeed expect the profile to develop and have a speed-up. Hence we calibrate the inflow by trial and error to reach the expected velocity magnitude and direction at the Tower 20, at a height of 100 m above the ground level. This is indeed the motivation to utilize a precursor developed inflow using a pressure-driven boundary layer and Coriolis forces. Weerasuriya *et al.* [13] showed that the wind veer (twist) of inflow profiles plays an important role in wind speed predictions for urban areas.

**Reviewer Point P 1.9** — Line 134: You write The direction of the flux automatically determines the inlet and outlet regions. I thought that the set wind direction would determine the inlet and outlet regions, or this is a misunderstanding from my side? How do you handle wind veer for determining the inlet and outlet boundaries?

**Reply**:  Yes, the direction of the mass flux, either from precursor or idealized profiles, determines whether a cell on the side boundary is set to be an inlet or an outlet [5].  This gives the same classification as to whether the inlet wind direction points inward or outward of the domain for most cases.  However, for some theoretical cases where the simulated flow is flowing back out of the cell, this cell is redefined as an outlet.  This is mainly avoided by choosing such an extensive domain with a smoothed outermost part of the terrain patch.

For the simulations using profiles obtained through the precursor, there will be wind veer such that not all points along a vertical straight line on the side patch are defined to be the same boundary type. This method has been validated on the OpenFOAM tutorials [10] that have been verified and validated for atmospheric boundary layer modelling on which we base our simulations.

**Reviewer Point P 1.10** — Section 3.2, Line 144 : What is the reason that you use a constant leaf area density? You could easily use a varying leaf area density based on the forest point cloud data.

**Reply**:  The objective of this study was to study the influence of canopy, and to test if the use of a canopy model could improve predictions close to the ground.  Simulations are planned with the use of a varying leaf area density based on the forest point cloud data as a next step. Indeed a more realistic representation of the forest, using different patches with different heights and removing the trees from the higher zones of the ridges could improve the results.

**Reviewer Point P 1.11** — Section 3.2, eq. (3): Is this really how the Coriolis force source term is implemented in your model? If you follow the ABL model of Koblitz et al. (2013) I would expect that you include $U$ and $V$-momentum source terms that represent a balance between a (constant) geostrophic wind speed and the Coriolis force (see for example van der Laan (2020)), as you also briefly discuss.

**Reply**:

As explained in Question 1, the Coriolis force source term is included as a momentum source term ($U$ and $V$ momentum equations), this has been corrected in the manuscript in the Methodology Section 3.2. The Coriolis force is calculated based on the planetary rotational period ($\Omega = 24\ h$) and the latitude ($\lambda$) for Perdigao (39.68 $°N$), where $f_c = 2\Omega sin(\lambda)$. Exactly similar to Koblitz et. al, the Coriolis force in vertical direction is neglected since it is small compared with the gravitational acceleration.

**Reviewer Point P 1.12** — Section 3.2, eqns (2) and (4) : I think you need to define different source terms, for example $S_{p,m}, S_{p,k}$ and $S_{p,\varepsilon}$.

**Reply**:  We thank the reviewer for the suggestion, $S_{p,m}$ is defined as the source term for the momentum equation, $S_{p,k}$ is defined as source term for turbulent kinetic energy and $S_{p,\varepsilon}$ is defined as source term for turbulent diffusion.  This description has been removed, as these terms are not activated for the neutral case.

**Reviewer Point P 1.13** — Table 2: The skipped entries ($-$""$-$) are unclear to me. For example, does the BBSF1 case include forest source terms or not? I would just fill in the entire table for clarity. In addition, what do you mean by using SKE for BBSF1? I though that you use a global length scale limiter in the epsilon equation, which is different from a standard model.

**Reply**:

Table 2, showing the different simulation case has been updated with all columns clearly filled. The BBSF model does not contain the canopy source term. The turbulence constants for the different models are further listed in a separate Table 3 in manuscript as shown below. Indeed, we use a global length scale limiter in the epsilon equation renamed as KE-Lim model.

Table 2: List of simulation cases simulating a period of neutral atmosphere, 22:00-22:30 (04.05.17).

| Case name | Solver | Inlet profiles | Source terms | | | | Turbulence mod. |
| | | | Canopy | Coriolis | Pres. gr. | Buoy. for. | |
| --- | --- | --- | --- | --- | --- | --- | --- |
| SF1 | SF | Idealized | No | No | No | No | SKE |
| SF2 | SF | Idealized | No | No | No | No | KO |
| SF3 | SF | Idealized | Yes | No | No | No | MKE |
| SF4 | SF | Idealized | No | Yes | No | No | SKE |
| BBSF | BBSF | Precursor | No | Yes | Yes | Yes | KE-Lim |

Table 3: Turbulence constants for different turbulence models

| Coefficient | Turbulence model | | | |
| | SKE | MKE | KO | KE-Lim |
| --- | --- | --- | --- | --- |
| $C_\mu$ | 0.09 | 0.033 | 0.09 | 0.09 |
| C1 | 1.44 | 1.44 | - | 1.44 |
| C2 | 1.92 | 1.92 | - | 1.92 |
| $\sigma_\epsilon$ | 1.30 | 1.85 | - | 1.30 |
| $\sigma_K$ | 1.0 | - | 0.5 | - |
| $\alpha_K$ | - | - | 0.5 | - |
| $\alpha_\omega$ | - | - | 0.6 | - |
| $\beta*$ | 0.09 | - | - | 0.09 |
| $\beta$ | - | - | 0.072 | - |
| $\nu$ | 1.5e-05 | 1.5e-05 | 1.5e-05 | 1.5e-05 |
| $L_{max}$ | - | - | - | 62.14 |
| $k_{Amb}$ | - | - | - | 0.001 |
| $\epsilon_{Amb}$ | - | - | - | $7.208e-08$ |
| $T_{Ref}$ | - | - | - | 300 |
| $Pr$ | - | - | - | 0.9 |
| $Prt$ | - | - | - | 0.74 |

**Reviewer Point P 1.14** — What do the error bars on the measurements represent? Is is the standard deviation of the uncertainty of the mean? I would recommend to use the latter.

**Reply**: Yes, the error bars represent one standard deviation of the mean measurements. This description has been updated in the manuscript in the introduction of Section 4 (Results). **Re-**

**viewer Point P 1.15** — You could group the profile plots in the results into three figures, where each row of sub plots represents a mast: SW ridge (combine Figs. 6 to 8), valley (combine Figs. 9 to 12), NE ridge (combine Figs. 13,14). I think this make sense because you also discuss them as met mast groups in the text. In addition, you could consider to plot normalized results of wind speed $U/U_{ref}$, turbulence intensity $\left(\sqrt{2/3k}/U_{ref}\right)$ and wind direction $(wd - wd_{ref})$ instead of dimensional results. Finally, you could zoom the $x$-axis of the wind direction, since it is hard to see the difference between the models and measurements in Figs. $6c, 7c, 8c$ and $14c$. This also applies to some of the wind speed and TKE plots.

**Reply**: We thank the reviewer for the suggestion. We have regrouped the profile plots in Figures 6-18 in terms of met-mast. We have also zoomed in the figures to clearly see the difference between the models and measurements.

**Reviewer Point P 1.16** — Line 194: You mention: A good match is obtained in between the measured and computed velocity, turbulent kinetic energy and wind direction profiles for the calibration Tower 20 as seen in Fig. 6. However, the results of the model including forest are not matching well, especially for the TKE and the wind speed near the ground.

**Reply**: We thank the reviewer for spotting this oversight. The sentence has been re-written in Section 4.1 of the manuscript.

**Reviewer Point P 1.17** — Figure 13: Nice plot. You could add the relevant mast location(s) in the plot, so the reader can better understand the results of Figs. 9-12.

**Reply**: We thank the reviewer for the suggestion. The nearest mast locations are now shown in Figure 15. Also, the locations of all the used met masts are presented in Figure 2. And a reference to this figure is added to the figure captions.

**Reviewer Point P 1.18** — I am missing information on code and data availability, which is normally added at the end of the article. In addition, I was wondering if is possible to provide the numerical setup using a DOI of a git hub repository through Zenodo or something similar. By proving the numerical setup/ run scripts, one could easily redo the work since the numerical solver (OpenFOAM) is publicly available.

**Reply**: A community has been setup in Zenodo (https://zenodo.org/communities/zephyr/). A repository of the numerical setup shall be added through Github (https://github.com/kartikv95/WESC-Perdigao).

**Minor**

**Reviewer Point P 1.19** — Introduction, Line 13: You could rewrite the following Lack of terrain availability in flat terrain pushes wind-farm developers to look for alternative sites along complex terrains., since you use the word terrain three times and I think complex terrains could be rewritten as complex terrain sites. The latter also applies elsewhere in the paper.

**Reply**: This has been modified. Thank you for the suggestion.

**Reviewer Point P 1.20** — Line 117: log-low → log-law.

**Reply**: This has been fixed. Thank you for the suggestion.

**Reviewer Point P 1.21** — There are quite a lot of other typos in the article but these can be fixed in the proof reading process.

**Reply**: The article has been proof-read. Thank you for the suggestion.

**References**

[1] Michael Alletto et al. "E-Wind: Steady state CFD approach for stratified flows used for site assessment at Enercon". In: *Journal of Physics: Conference Series* 1037 (June 2018), p. 072020. DOI: 10.1088/1742-6596/1037/7/072020.

[2] David Apsley and Ian Castro. "A Limited-Length-Scale k − $\epsilon$ Model for the Neutral and Stably Stratified Atmospheric Boundary Layer". In: *Boundary-Layer Meteorology* 83 (Apr. 1997), pp. 75–98. DOI: 10.1023/A:1000252210512.

[3] Clara Garcia-Sanchez et al. "Inflow Uncertainty Quantification of dispersion in Oklahoma City". In: June 2015.

[4] M. Gaumond et al. "Evaluation of the wind direction uncertainty and its impact on wake modeling at the Horns Rev offshore wind farm". In: *Wind Energy* 17 (Aug. 2014). DOI: 10.1002/we.1625.

[5] *Inlet Outlet Boundary condition*. URL: https://www.openfoam.com/documentation/guides/latest/doc/guide-bcs-inlet-atm-atmBoundaryLayer.html.

[6] M. Paul van der Laan, M. Baungaard, and Mark Kelly. "Inflow modeling for wind farm flows in RANS". In: *Journal of Physics: Conference Series* 1934 (May 2021), p. 012012. DOI: 10.1088/1742-6596/1934/1/012012.

[7] M. Paul van der Laan, Mark Kelly, and Mads Baungaard. "A pressure-driven atmospheric boundary layer model satisfying Rossby and Reynolds number similarity". In: *Wind Energy Science* 6 (June 2021), pp. 777–790. DOI: 10.5194/wes-6-777-2021.

[8] M. Paul van der Laan et al. "The k-$\epsilon$-fP model applied to wind farms". In: *Wind Energy* 18 (Sept. 2014). DOI: 10.1002/we.1804.

[9] Jose Laginha Palma et al. "The digital terrain model in the computational modelling of the flow over the Perdigão site: the appropriate grid size". In: *Wind Energy Science* 5 (Nov. 2020), pp. 1469–1485. DOI: 10.5194/wes-5-1469-2020.

[10] *OpenFOAM Atmospheric models verification and validation*. URL: https://develop.openfoam.com/Development/openfoam/-/tree/master/tutorials/verificationAndValidation/atmosphericModels.

[11] Jonas Schmidt, Carlos Peralta, and Bernhard Stoevesandt. "Automated generation of structured meshes for wind energy applications". In: London: Open Source CFD International Conference, London, Oct. 2012.

[12] Lieven Vervecken, Johan Camps, and Johan Meyers. "Accounting for wind-direction fluctuations in Reynolds-averaged simulation of near-range atmospheric dispersion". In: *Atmospheric Environment* 72 (2013), pp. 142–150.

[13] A.U. Weerasuriya et al. "Integrating twisted wind profiles to Air Ventilation Assessment (AVA): The current status". In: *Building and Environment* 135 (2018), pp. 297–307. ISSN: 0360-1323. DOI: https://doi.org/10.1016/j.buildenv.2018.03.024. URL: https://www.sciencedirect.com/science/article/pii/S0360132318301537.

---

## Author Comment (AC2)

**Response to the reviewer**

We thank the reviewer #2 for their useful comments and the time invested in reviewing our manuscript. We have addressed each of the referee comments as detailed point by point below, which we believe has significantly improved the quality of the manuscript.
* * *
**Reviewer 1**

**Main Comments**

**Reviewer Point P 1.1** — From reading the article, one can not know precisely what TKE models parameters were used in the simulations. As an example, the $k - \varepsilon$ model can use different model constant values such as $C_\mu$ used in the calculation of the turbulent viscosity $\mu_T = \rho C_\mu \frac{k^2}{\varepsilon}$. Is is usually used the standard value $C_\mu = 0.090$; however, for atmospheric flows $C_\mu = 0.033$ seems to be more adequate. Is that the case? There are also other parameters that are not revealed, such as the Prandtl number for the turbulent dissipation and others. Do you use the default values in OpenFOAM? Even if it is the case, it should be enumerated.

**Reply**: Indeed, this is an important point, we thank the reviewer for highlighting this missing information. All the TKE models parameters that were used in the simulations are now enumerated in Table 3 in the manuscript. For the Canopy model, the $k - \epsilon$ turbulence model utilized had a coefficient $C_\mu = 0.033$, however, for the other models the default coefficient of $C_\mu = 0.09$ was utilized. Other constant values such as the Prandtl number for the turbulent dissipation are also defined in Table 3.

Table 1: Turbulence constants for different turbulence models

| Coefficient | Turbulence model | | | |
|---|---|---|---|---|
| | SKE | MKE | KO | KE-Lim |
| $C_\mu$ | 0.09 | 0.033 | 0.09 | 0.09 |
| C1 | 1.44 | 1.44 | - | 1.44 |
| C2 | 1.92 | 1.92 | - | 1.92 |
| $\sigma_\epsilon$ | 1.30 | 1.85 | - | 1.30 |
| $\sigma_K$ | 1.0 | - | 0.5 | - |
| $\alpha_K$ | - | | 0.5 | - |
| $\alpha_\omega$ | - | - | 0.6 | - |
| $\beta*$ | 0.09 | - | - | 0.09 |
| $\beta$ | - | - | 0.072 | - |
| $\nu$ | 1.5e-05 | 1.5e-05 | 1.5e-05 | 1.5e-05 |
| $L_{max}$ | - | - | - | 62.14 |
| $k_{Amb}$ | - | - | - | 0.001 |
| $\epsilon_{Amb}$ | - | - | - | $7.208e-08$ |
| $T_{Ref}$ | - | - | - | 300 |
| $Pr$ | - | - | - | 0.9 |
| $Prt$ | - | - | - | 0.74 |

**Reviewer Point P 1.2** — A similar problem appear in the canopy model description; the model

parameters aren't completely defined. It is said that the model is based in [Lopes da Costa, 2007], but this model uses several parameters that are not fully addressed in this article. It is presented the source term for the velocity and one can not know what is the $\alpha$ parameter in it or what that represents. The source/ sink terms for the turbulent kinetic energy and its dissipation rate are not also mentioned. I think that you might be more clear in this subject.

**Reply**: Indeed, we thank the reviewer for pointing out this missing information. The canopy model description has been updated in the revised manuscript. We apologize for the inconsistency in the parameter definitions, this has now been corrected throughout the document. The canopy source term was utilized with the simpleFoam flow solver. The porosity model is based on the"powerLawLopesdaCosta" model implemented in OpenFOAM [1]. It is a variant of the power law porosity model with spatially varying drag coefficient. This source term is applied to the momentum equation to reproduce the momentum dissipation that the trees and its foliage should produce in the flow. The following parameters are used: Porosity surface area per unit volume or the leaf area density ($\Sigma = 1.0$), Drag coefficient ($C_d = 0.25$), and the Power law model exponent coefficient ($C1 = 2.0$). The $k - \epsilon$ turbulence model utilized with the canopy model uses the following model constants as enumerated in Table 3 in the manuscript. These details have been added in the Methodology section of the manuscript (Section 3.2).

| Constant | Value |
|---|---|
| $C_{mu}$ | 0.033 |
| C1 | 1.44 |
| C2 | 1.92 |
| $\sigma_{eps}$ | 1.85 |

Alternatively, there are canopy source terms that can be utilized buoyantBoussinesqSimpleFoam solver. These included the atmPlantCanopyUSource, atmPlantCanopyTSource, atmPlantCanopyTurbSource for momentum, temperature and turbulence respectively, taking into account the thermal and turbulence effects induced the canopy into account. The source/ sink terms for the turbulent kinetic energy and its dissipation rate have not been utilized for the present study on a complex terrain, as we encountered convergence issues that needs to be resolved.

**Reviewer Point P 1.3** — In chapter 3 , though the definition of the domain volume is well explained, the description of the computational mesh is vague. It is said that it consists in 12.7 million cells, but it would be more useful to define it by the number of cells per main direction, such as $n_x \times n_y \times n_z$. It is said that the horizontal mesh resolution was set to 33 m, but is this resolution observed only in the center of the domain (close to the masts location) expanding itself to the boundaries or is it a regular mesh? (...which is probably not the case, as the domain is not square or rectangular, but cylindrical.) It is also said that an "uniform stretching is applied to the vertical direction" with no more details. However, one of the most important mesh parameters in an atmospheric flow simulation is the minimal mesh height $\Delta z$ next to the ground (along with the vertical number of cells), which is not presented in the article.

**Reply**: Indeed, we apologize for the incomplete description of the computational mesh, it is now specified as follows in the revised manuscript. The overall mesh consists of 12.7 million cells. In terms of the number of cells per main direction ($N_x \times N_y \times N_z$) the mesh comprises of 227 x 227 x 120 across the terrain patch. The minimal mesh height $\Delta z$ next to the ground is close to 3 m. The vertical mesh resolution is 33 m, with a stretching factor applied to cluster cells close to the ground as seen in Fig 4 in

the manuscript. The terrainBlockmesher [3] tool uses a blending function to smooth the transition from the terrain patch to the outer cylindrical block. Around 50 radial block cells are defined and a radial grading factor is used to enable a stretching in the horizontal direction to cluster cells across the centre of the domain (close to met mast locations) and expanding towards the boundaries. This description has been updated in the Methodology section of the manuscript (Page 5).

**Reviewer Point P 1.4** — It is said (line 145) that you have choose a mean tree height of 3 m. I suppose that it is all over the domain (or, at least in the $7, 5$ km$\times$ 7.5 km squared area - it is not clear in the article. Wouldn't it be more correct to use different patches (eventually with different heights), as you have that information in figure 5? Or, at least remove the trees from the higher zones of the ridges, as it is common to happen in this kind of topography and cam be observed in figure 5?

**Reply**:
    We agree with the reviewer on this point. It would be more correct to use different patches with different heights as seen in the forest point cloud data. We choose a mean tree height of 3 m applied all over the domain. A cell set was utilized to select a volume of cells 3 m above the ground (this forms a canopy zone where the source terms are activated). This description has been updated in the article in Section 3.2. Indeed this can be modified to be more specific in the choice of areas to activate the canopy model. This requires a use of a function that interpolates data from the forest point cloud for selection of canopy height for each point on the terrain patch, which is presently being coded.

**Reviewer Point P 1.5** — I suppose that you use in the canopy model an uniform area density. However, forests have a higher foliage density at the half top than at the bottom zones - [Lalic and Mihailovic, 2004]. I think a non-uniform leaf area density could be easily implemented in that model.

**Reply**:
    We thank the reviewer for this suggestion. Indeed, as highlighted by Lalic and Mihailovic [2], forests have a higher foliage density at the half top than at the bottom zones. The objective of this study was to study the influence of canopy, and to test if the use of a canopy model could improve predictions close to the ground with uniform area density. Future simulations can be performed with a non-uniform leaf area density based on an analysis of the foliage composition in Perdigao [4].

**Reviewer Point P 1.6** — I find the Figure 13 very interesting. The extension of the slice could be wider, in order to include the whole top of the ridges and clearly show the positions of the masts 10 and 7 (and approximately the zone of mast 37 ). This small change could complement and be more enlightening in the interpretation of the results obtained for these masts and presented in the other figures. (Also could be useful for the reader if the caption of the figure refer that the location of slice is defined in Figure 2.)

**Reply**: Thank you for this very valuable suggestion, which we have taken into account by updating Figure 15 and also Figure 2 accordingly. The captions have also been updated and other relevant captions, referring to the location of the slice and masts 10, and 7 and (approximately the zone of mast 37) in Figure 2.

**Minor**

**Reviewer Point P 1.7** — Line 111: ”..homogeneous atmospheric boundary layer (ABL). either...” should be ”...homogeneous atmospheric boundary layer (ABL) either...”.

**Reply**: This typo has been correct in the manuscript. Thank you for the suggestion.

**Reviewer Point P 1.8** — Figure 5: The scale in the figure is strange; the sequence of the $d($ m$)$ values is $0, 2, 6, 4, 8$; shouldn't it be $0, 2, 4, 6, 8$? 3. For example: In Figure 5, $m$ for meters shouldn't be written in italic as it is not a variable, but a length unit. In many other figures there is also the variables $(U, TKE, ...)$ not in italic, but with the units in italic; it should be the opposite.

**Reply**: Thank you for the suggestion. Yes, the sequence of the bar values was wrong an have now been fixed. Figures with variables and units in wrong text style regarding italic/non-italic has been fixed. It includes Fig. 2, Fig. 3, Fig.5. 6-14 and 16-22

**Reviewer Point P 1.9** — Table 2: I would prefer the description of each case to be done in a more clear way, instead of using ”-”- ” all over the table.

**Reply**: Table 2 has been fixed in the manuscript with all the columns clearly written. Thank you for the suggestion

**Reviewer Point P 1.10** — Line 111: ”predictions” is two times written.

**Reply**: The typo has been fixed. Thanks for the suggestion.

**Reviewer Point P 1.11** — Line 205 to 206 : I think that the order of the towers and the figures is somehow messed up. It should be ”...towers $25, 7, 27$ and $22$.”, and further, ”...shown in Figs. 9, 10, 11 and 12 respectively.”.

**Reply**: This has been fixed. The figures have been regrouped in terms of group of masts for a specific variable.

**Reviewer Point P 1.12** — ...and a small detail: in the references, ”Costa, J. L. C. (2007)” should be ”Costa, J. C. L. (2007) ”, or even ”Lopes da Costa, J. C. (2007)″. ; )

**Reply**: This has been fixed. We apologize for the oversight.

**References**

[1] *API guide: powerlawlopesdacosta class reference.* URL: https://www.openfoam.com/documentation/guides/latest/api/classFoam_1_1porosityModels_1_1powerLawLopesdaCosta.html.

[2] Branislava Lalic and Dragutin Mihailovic. "An Empirical Relation Describing Leaf-Area Density inside the Forest for Environmental Modeling". In: *Journal of Applied Meteorology - J APPL METEOROL* 43 (Apr. 2004), pp. 641–645. DOI: 10.1175/1520-0450(2004)043<0641:AERDLD>2.0.CO;2.

[3] Jonas Schmidt, Carlos Peralta, and Bernhard Stoevesandt. "Automated generation of structured meshes for wind energy applications". In: London: Open Source CFD International Conference, London, Oct. 2012.

[4] C.A.M. Silva et al. "Surface cover in Perdigão: forest delineation - 2nd Workshop on Perdigão". In: Mar. 2019.

---

## Referee Report (RR1)

**Review of R1: Effect of different source terms in atmospheric boundary modelling over the complex terrain site of Perdigao by K. Venkatraman et al.**

Reviewer: M. Paul van der Laan, DTU Wind Energy

June 8, 2022

I would like to thanks the authors for their answers and corrections. I have a number of remaining comments that are not properly addressed. My main concern is related to the grid refinement study, which shows that the applied grid is not sufficient. Therefore, the conclusions of the article do no hold and I cannot recommend the article to be accepted in the present form. The grid refinement study itself is neither sufficient, as described below.

**Main comments**

- 1. P 1.1: There are still undefined parameters in the article: for example what is the value of G?
- 2. P 1.5: It is great that you have added a grid refinement study. What is the reason that you have not changed the number of cells in the vertical direction? It seems that you have only looked at the influence of the horizontal grid refinement. A proper grid refinement study needs to include all three directions. In addition, the grid refinement study shows some worrying results of the wind speed profiles because the results are not converging with grid refinement (the difference between medium and fine is larger than the difference between coarse and medium. This indicates that you need a finer grid or there is something wrong with the numerical setup. Furthermore, the grid refinement study indicates that your current chosen grid size (the coarsest grid in the grid refinement study) is not sufficient. This is major problem in the article because all the conclusions are based on the results of the coarsest grid.
- 3. P1.6: Great that you have added results for the inflow profiles. It seems that the ABL setup (with Coriolis and ABL height) under predicts the TKE (or TI) by quite a margin, which makes a comparison/validation with measurements challenging. You could actually find a set of G and  $\ell_{\text{max}}$  that gives you a matching TI at a reference height (as long as the TI value exist for a given  $z_0$  and G). See for example the Appendix of a recent work of my own [1]. If you cannot get a matching TI, then you could choose to change the roughness height for the ABL inflow.
- 4. P1.7: You mention that you focus on the influence of the source terms; however, you do validate and evaluate the performance of each the model with the measurements throughout the article and in the conclusion. Hence, I think it makes sense to perform a range of wind wind directions and apply a Gaussian filter as post processing step, especially if the wind direction standard deviation is as large as 7°. You could at least perform two additional wind directions representing the standard deviation  $(231 \pm 7^{\circ})$  and look at the difference between the three wind directions.

**References**

 van der Laan, M. P., Kelly, M., and Baungaard, M. A pressure-driven atmospheric boundary layer model satisfying rossby and reynolds number similarity. *Wind Energy Science*, 6(3):777– 790, 2021.

---

## Referee Report (RR2)

**Review of *R2: Effect of different source terms and inflow direction in atmospheric boundary modeling over the complex terrain site of Perdigao* by K. Venkatraman et al.**

Reviewer: M. Paul van der Laan, DTU Wind Energy

September 26, 2022

I would like to thanks the authors for their answers and corrections. It is nice that you have added the different wind direction results; it highlights the challenge of complex terrain modeling and validation with measurements that include a varying inflow direction.

Unfortunately, the revised grid refinement study is not properly performed. As this already the third round of review I would suggest to either add a proper grid refinement study or simply remove the grid refinement study and write that a proper grid refinement study is future work, and that the results presented could change if a finer resolution is used. Hereby some more detailed info:

**Main comments**

1. The revised grid refinement study uses a grid size that hardly changes, namely, $N_x = 550, 600, 650$, which is a refinement ratio of 1.08-1.09; the previous study used $N_x = 227, 332, 469$. This revised grid refinement study uses far too small refinement ratios and it is likely that you get similar results between the different grids as you also find; you write at Line 109: *The results obtained with 3 different meshes of increasing resolution show negligible sensitivity on the wind profiles at three different towers on the ridges and inside the valley.* In this case, you cannot conclude to get grid independent results. You need to have at least a grid refinement ratio of $\sqrt{2}$, similar to what you used in R1 (personally I always use a factor 2). You mention that you are limited to memory requirements, but you can always add results of coarser grids, for example you could use something as $N_x = 300, 450, 675$ using a refinement ratio of 1.5. If the grid results indicate that the error due to grid resolution is not converging then you would need to go finer. If you are limited to memory to do so, then you could also change to a higher order numerical scheme to reduce the grid resolution errors or simply run on a high performance computer cluster, which I believe the von Karman institute has access to (https://www.vki.ac.be/index.php/facilities-other-menu-148/hpc-cluster)

   There is also a choice made in the revised grid refinement study that I would have done differently, which I would like to share for food for thought. When I perform a grid refinement study then I would not change the inflow profile per grid size because it would mean that you would model a different case per grid size. I do understand your choice because you focus on validation and are trying to mimic the inflow conditions of the measurements. However, for a more fair grid refinement study it would be better to not change the inflow parameters in my opinion and separate the model verification (grid refinement study) from the model validation (comparison with measurements). For example, you could have chosen to use the inflow profile based on the finest grid and use this inflow for all other grid sizes. Furthermore, the results of a grid refinement does not have to be compared with measurements as the reader could then be tempted to pick a grid result that is closest to the measurements, instead of taking the grid size that has negligible numerical errors due to grid resolution; the latter is the purpose of a grid refinement study.

---

## Author Response (AR2)

**Response to the reviewer**

We thank the reviewer #1 for their useful comments and the time invested in reviewing our manuscript. We have addressed each of the referee comments as detailed point by point below, which we believe has significantly improved the quality of the manuscript.

**Reviewer 1**

**Main Comments**

**Reviewer Point P 1.1** — P 1.1: There are still undefined parameters in the article: for example what is the value of G?

**Reply:**

We thank the reviewer for this point. The undefined parameters have been updated in Table 3. The geostropic wind speed (G) is set to around 6.2 m/s, to calibrate the model profile at 573 m, to obtain the required calibration wind speed at a tower height of 100 m.

**Reviewer Point P1.2** — P 1.5: It is great that you have added a grid refinement study. What is the reason that you have not changed the number of cells in the vertical direction? It seems that you have only looked at the influence of the horizontal grid refinement. A proper grid refinement study needs to include all three directions. In addition, the grid refinement study shows some worrying results of the wind speed profiles because the results are not converging with grid refinement (the difference between medium and fine is larger than the difference between coarse and medium. This indicates that you need a finer grid or there is something wrong with the numerical setup. Furthermore, the grid refinement study indicates that your current chosen grid size (the coarsest grid in the grid refinement study) is not sufficient. This is major problem in the article because all the conclusions are based on the results of the coarsest grid.

**Reply:**

The grid refinement study has been performed, with three different mesh cases and changing the number of cells in all three directions. All the cases that have been run are finer than any case previously used. Previously, due to computational and memory requirements, we had a limit on the number of cells and kept a cap on the number of cells in the vertical direction. The structured grids are generated using the terrainBlockMesher tool [2], which interpolates the SRTM terrain data and creates the terrain patch which is blended into a cylindrical domain. The simulation inflow profiles for all the meshes are calibrated to match the field experiment at 100 m of Tower 20 (tse04) on the South-West ridge. A convergence of the wind profiles is seen at towers across the towers on the South-West ridge, inside the valley and on the North East ridge. The simpleFoam (SF1) solver with the  $k - \epsilon$  turbulence model setup is utilized for the study. The medium case is chosen for the present mesh study and all the simulations with different models have been re-performed. The mesh resolution cases are well within the recommendations provided by Palma *et al.* [1].

| Case   | Nx  | Ny  | $\mathbf{N}\mathbf{z}$ | NCells (million) |
|--------|-----|-----|------------------------|------------------|
| Coarse | 550 | 550 | 150                    | 62               |
| Medium | 600 | 600 | 170                    | 88               |
| Fine   | 650 | 650 | 190                    | 115              |

Table 1: Grid refinement study parameters showing the number of cells per main direction.

Figure 1: a) Wind velocity magnitude b) Wind direction c) Turbulent kinetic energy tuned to reach calibration at height 573 m corresponding to 100 m at Tower 20 (tse04)

Figure 2: a) Wind velocity magnitude b) Wind direction c) Turbulent kinetic energy at Tower 25 (tse09)

---

## Author Response (AR3)

**Response to the reviewer**

We thank the reviewer #1 for their useful comments and the time invested in reviewing our manuscript. We have addressed each of the referee comments as detailed point by point below, which we believe has significantly improved the quality of the manuscript.

**Reviewer 1**

**Main Comments**

**Reviewer Point P1.1 —**

The revised grid refinement study uses a grid size that hardly changes, namely,  $N_x = 550, 600, 650$ , which is a refinement ratio of 1.08-1.09; the previous study used  $N_x = 227, 332, 469$ . This revised grid refinement study uses far too small refinement ratios and it is likely that you get similar results between the different grids as you also find; you write at Line 109: The results obtained with 3 different meshes of increasing resolution show negligible sensitivity on the wind profiles at three different towers on the ridges and inside the valley. In this case, you cannot conclude to get grid independent results. You need to have at least a grid refinement ratio of  $\sqrt{2}$ , similar to what you used in R1 (personally I always use a factor 2). You mention that you are limited to memory requirements, but you can always add results of coarser grids, for example you could use something as  $N_x = 300, 450, 675$  using a refinement ratio of 1.5. If the grid results indicate that the error due to grid resolution is not converging then you would need to go finer. If you are limited to memory to do so, then you could also change to a higher order numerical scheme to reduce the grid resolution errors or simply run on a high performance computer cluster, which I believe the von Karman institute has access to (https://www.vki.ac.be/index.php/facilities-other-menu-148/hpc-cluster)

There is also a choice made in the revised grid refinement study that I would have done differently, which I would like to share for food for thought. When I perform a grid refinement study then I would not change the inflow profile per grid size because it would mean that you would model a different case per grid size. I do understand your choice because you focus on validation and are trying to mimic the inflow conditions of the measurements. However, for a more fair grid refinement study it would be better to not change the inflow parameters in my opinion and separate the model verification (grid refinement study) from the model validation (comparison with measurements). For example, you could have chosen to use the inflow profile based on the finest grid and use this inflow for all other grid sizes. Furthermore, the results of a grid refinement does not have to be compared with measurements as the reader could then be tempted to pick a grid result that is closest to the measurements, instead of taking the grid size that has negligible numerical errors due to grid resolution; the latter is the purpose of a grid refinement study.

**Reply**: We thank the reviewer for this comment. As indicated, the grid refinement study uses small refinement ratios and it is indeed likely that there are similar results between the different grids due to this reason. For the updated study, we use a grid coarsening approach in all three directions. Two additional coarse grids ("Very Coarse" and "Coarse" cases) with a grid refinement factor of around 1.4 are added for the mesh refinement study along with the previous three grids as shown in Table 1. Based on the study we conclude that the present results indeed appear to be grid-dependent for the range of grids that have been investigated and a further study needs to be done on a finer grid which is a part of future work. This sentence has been added to the paper in the Methodology and Appendix sections.

| Case        | $\mathbf{N}\mathbf{x}$ | Ny  | $\mathbf{Nz}$ | NCells (million) |
|-------------|------------------------|-----|---------------|------------------|
| Very Coarse | 300                    | 300 | 80            | 12               |
| Coarse      | 460                    | 460 | 120           | 30               |
| Medium      | 500                    | 500 | 150           | 62               |
| Fine*       | 550                    | 550 | 170           | 88               |
| Very Fine   | 650                    | 650 | 190           | 115              |

Table 1: Grid refinement study parameters showing the number of cells per main direction.

Significant differences are seen between the very coarse and medium grids especially close to the ground near the Tower 20 (Calibration tower) on the South West ridge seen in Fig 1, inside the valley as seen in Fig. 2, and at Tower 29 close to 100 m on the North East ridge seen in Fig 3. Larger differences are seen close to the ground as the topology of the terrain is further resolved near the surface upon grid refinement. The structured grids are generated using the terrainBlockMesher tool [1], which interpolates the SRTM terrain data and creates the terrain patch on which smoothing is applied towards a cylindrical domain on the sides. Increasing the number of cells refines and slightly modifies the surface mesh close to the ground. Furthermore, a small change in the prediction of the extent of the re-circulation zone could have a significant change and uncertainty in the predictions inside the valley and on top of the Northeast ridge. This suggests that grid spacing is an important parameter for flow prediction around a complex terrain, and a grid-independent solution could be challenging to achieve based on the complexity of the flow topology. The difference in profiles is of the same order of magnitude as the differences caused by different model setups.

We also plan to include this grid study in the Appendix as the editor suggested investigating if the "grid refinement error" is of the same order as the differences caused by other set-up selections, in order to identify the signals that potentially dominate the simulation error when compared to experiments and mentioning it as future work for performing simulations on a much finer mesh. As a clarification, we indeed run simulations on the cluster. However, we generate the mesh on a local/personal machine, since the grid generation is compiled for an older version of OpenFOAM and the memory issues are related to the creation of the mesh. The "Very Fine" grid mesh is the maximum number of cells that could be generated. Hence we suggest that future work should include an investigation with a much smaller grid size. We also agree that the best way is not to change the inflow for each grid. However, we were not sure of deciding on the finest grid and the inflow applied to it, hence we opted for the calibration approach for each grid.

Figure 1: a) Wind velocity magnitude b) Wind direction c) Turbulent kinetic energy tuned to reach calibration at height 573 m corresponding to 100 m at Tower 20 (tse04)

Figure 2: a) Wind velocity magnitude b) Wind direction c) Turbulent kinetic energy at Tower 25 (tse09)

---

## Author Response (AR4)

**Response to the editor**

We thank the editor for the useful comments and suggested corrections.

1. In the references, Vasiljevic et al and Wagner et al are in WESD. Please refer to the final published paper in WES.

2. In the Koblitz reference, please refer also to the university where the thesis is published.

3. There are several titles of the papers where capitalisation is missing ("bolund"-"Bolund", "perdigao"-"Perdigao", ...., Use Bolund in LaTeX).

4. Since you indicate that the extent of the recirculation zones is critical, I think it is worthwhile to refer to https://acp.copernicus.org/articles/19/2713/2019/ where these are measured under various stability conditions.

**Reply**: The above corrections have been made in the manuscript. The references have been fixed and also the reference to the article on re-circulation zones highlighted in the link has been added.